# AIM: ADVERSARIAL INFORMATION MASKING FOR EVALUATING EEG-DL INTERPRETATIONS

## ABSTRACT

We identify significant gaps in the existing frameworks for assessing the faithfulness of post-hoc explanation methods, which are essential for interpreting model behavior. To overcome these challenges, we propose a novel adversarial information masking (AIM) approach that enhances in-distribution information masking techniques. Our study conducts the first quantitative comparison of faithfulness assessment frameworks across different architectures, datasets, and domains, facilitating a comprehensive evaluation of post-hoc explanation methods for deep learning of human electroencephalographic (EEG) data. This work lays a foundation for further developments of reliable applications of explainable artificial intelligence (XAI). The code and sample data for this work are available at https://anonymous.4open.science/r/EEG-explanation-faithfulness-5C05.

## 1 INTRODUCTION

Recent advances in deep learning (DL) have strengthened the discussion around eXplainable Artificial Intelligence (XAI) (Zeiler & Fergus, 2014; Samek et al., 2016; Lundberg & Lee, 2017). Since most deep neural networks operate as "black boxes" that lack direct interpretability (Samek et al., 2016; Ancona et al., 2017), XAI is essential for three main reasons. First, it enables effective evaluation of AI-assisted decision-making processes (Goodman & Flaxman, 2017). Second, it assists researchers in debugging and improving DL models (Cadamuro et al., 2016; Adebayo et al., 2020; Krishna et al., 2024). Third, XAI reveals information that may be hidden from human perception (Shrikumar et al., 2017). Among various XAI methods, model-agnostic approaches that provide insight into what a model has learned are referred to as post-hoc explanations. These explanations can be categorized based on the level of features they address, ranging from human-interpretable high-level representations to low-level input features, with the categories termed training point ranking, concept activation, and feature attribution (Adebayo et al., 2022). In the field of electroencephalogram (EEG) analysis, the use of DL for decoding task-related patterns has shown significant success, leading to increased interest in recent years (Roy et al., 2019). In EEG-DL research, feature attribution methods enhances our understanding of both the EEG data and the models employed through visualizing saliency of input features (Tjoa & Guan, 2020; Pan et al., 2022; Bilodeau et al., 2024).

As the number of feature attribution methods expands, new concerns emerge regarding the quality of the explanations generated. Quantitative assessment of explanation quality remains a challenge, as it is often difficult to differentiate between model misbehavior and flaws inherent in the attribution methods (Sundararajan et al., 2017). Nonetheless, criteria for evaluating quality have become more widespread over the past decade. We summarize the related research in Table 1, which provides a general overview of the current landscape. Recent efforts have primarily focused on assessing the effectiveness of explanations in accurately representing the features that significantly influence model decisions, a quality we will refer to as "faithfulness". As Shah et al. (2021) suggested, a larger feature attribution indicate a higher relevance to model prediction. This can be understood in two key ways: 1) model decisions can reflect the presence or absence of an input feature, and 2) perturbations to important features tend to have a more pronounced impact on model decisions, and vice versa.

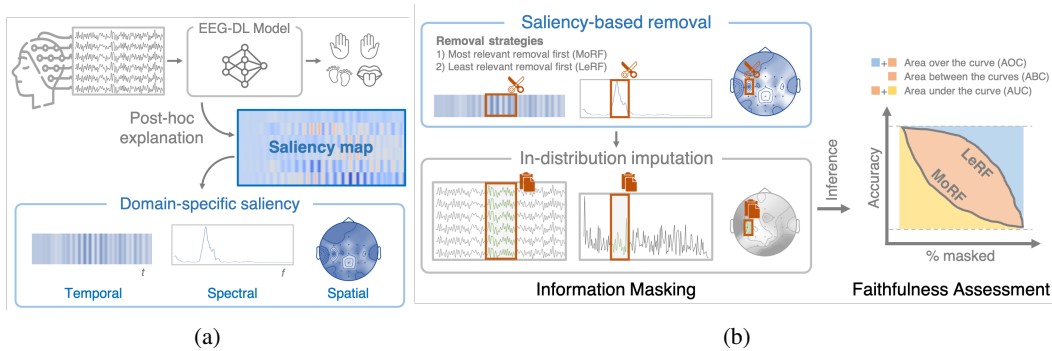

(a)                 (b)

Figure 1: (a) In the context of an EEG-DL recognition, post-hoc explanations provide feature attribution scores that are visualized as saliency maps. These saliency maps allow for the extraction of domain-specific saliency, facilitating interpretations across the temporal, spectral, and spatial domains. (b) To quantify the faithfulness of a feature attribution method based on generated saliency maps, information masking removes features following the Most Relevant Features (MoRF) and Least Relevant Features (LeRF) strategies and imputes them with in-distribution data. The model's inference accuracy is then evaluated against the masking ratio. Finally, we assess the faithfulness of the feature attribution method using metrics based on the areas over the MoRF curve, under the LeRF curve, and between the two curves.

Building on the two key points, we categorize existing evaluation strategies for faithfulness into two main types: "*fidelity analysis*" (Yeh et al., 2019) and "*robustness analysis*" (Hsieh et al., 2020; Fang et al., 2024). *Fidelity analysis* quantifies the discrepancy between perturbations of input features based on explanations and the expected changes in model output. Since the absence of an important feature should result in a significant decline in model performance, a smaller discrepancy between actual degradation and expected change reflects higher fidelity or greater faithfulness of the explanation. Conversely, *robustness analysis* examines whether attribution magnitudes are positively correlated with a feature's susceptibility to adversarial attacks. Features that are less influential to model decisions should show greater tolerance to adversarial perturbations, and a faithful explanation should accurately represent this by identifying such features as unimportant.

However, existing frameworks for assessing faithfulness face several challenges:

- Current evaluations of post-hoc explanation methods rely on suboptimal information masking techniques, which can result in out-of-distribution imputations when applied to real-world data.

- Although multiple frameworks have been proposed, there is no standardized methodology that enables a quantitative comparison among these explanation methods.

The challenges are elaborated in Section 2.1. To address these gaps in the context of EEG-DL analysis, given the number of studies that have adopted XAI (Tjoa & Guan, 2020; Sujatha Ravindran & Contreras-Vidal, 2023), our study proposes comprehensive faithfulness evaluation frameworks incorporating multi-domain information masking techniques.

Our primary objective is to determine which explanation methods are most suitable for elucidating EEG-DL models. Our contributions include:

1. We expand the leading in-distribution information masking method, Remove and Debias, to accommodate multiple domains, including spatial, temporal, and spectral dimensions.

2. We introduce an adversarial information masking (AIM) approach to circumvent issues related to hand-crafted distribution selection and to enhance in-distribution information masking for multivariate time series data.

3. We assess the effectiveness of in-distribution information masking through a novel Multi-Domain Adversarial Robustness (mdAR) framework that includes new normalized faithfulness metrics and an evaluation result consistency-based methodology for framework validation.

4. We demonstrate assessments of faithfulness for existing post-hoc explanation methods and their limitations under specific conditions in the context of deep learning interpretation of human EEG data.

Table 1: Obfuscating articulations of homogeneous explanation quality criteria in referenced studies. This study focuses on the evaluation of *explanation faithfulness*.

| Terminology | Study | Removal/Imputation | Motifs |
|---|---|---|---|
| Sensitivity-n | Ancona et al. (2017) | – | |
| Completeness | Sundararajan et al. (2017) | – | |
| Sensitivity (a), (b) | Sundararajan et al. (2017) | – | *Explanation axioms*: Mathematical property or |
| Linearity | Sundararajan et al. (2017) | – | quantitative relationship with input information that the |
| Summation to delta | Shrikumar et al. (2017) | – | attributed saliency values should satisfy. |
| Local Accuracy | Lundberg & Lee (2017) | – | |
| Missingness | Lundberg & Lee (2017) | – | |
| Sensitivity | Kindermans et al. (2019) | – | *Explanation robustness*: How easy it is to distort or |
| Similarity | Adebayo et al. (2018) | – | manipulate attribution result, or the variance of |
| Sensitivity | Yeh et al. (2019) | – | attributed saliency pattern under fundamentally similar |
| Robustness | Sujatha Ravindran & Contreras-Vidal (2023) | – | generation settings. |
| Quality | Samek et al. (2016) | Remove | |
| Consistency | Lundberg & Lee (2017) | – | |
| Fidelity | Yeh et al. (2019) | Remove | |
| Fidelity | Tomsett et al. (2020) | Remove | |
| Fidelity | Brocki & Chung (2022) | Remove | |
| Sensitivity | Cui et al. (2023) | Remove | *Explanation faithfulness*: Genuity of saliency with |
| Importance Accuracy | Hooker et al. (2018) | ROAR | regard to model decision. In most cases contrives a |
| Fidelity / Faithfulness | Shah et al. (2021) | DiffROAR | strategy to handle the interaction of explanation and |
| Fidelity | Rong et al. (2022) | ROAD | model decision. |
| Reliability | Torres et al. (2023) | ROAR | |
| Importance Accuracy | Park et al. (2023) | GOAR | |
| Effectiveness | Turbé et al. (2023) | Corrupt and train | |
| Quality | Hsieh et al. (2020) | AR | |
| Faithfulness | Fang et al. (2024) | OAR | |
| Sensitivity | Sujatha Ravindran & Contreras-Vidal (2023) | Noise Ratio | |

## 2 RELATED WORK

In this section, we identify potential issues within evaluation frameworks, provide a synthesis of key benchmarking approaches, and offer a concise overview of explanation evaluation in EEG analysis. Given the diverse terminology employed across the literature, our focus is on accurately conveying the fundamental concepts rather than adhering rigidly to the specific language used in individual references.

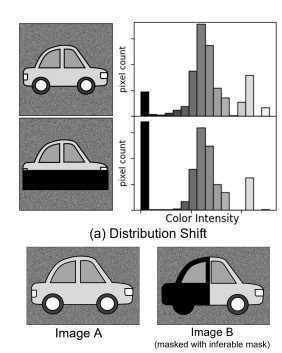

(a) Distribution Shift

Image A     Image B
(masked with inferable mask)
(b) Information Leakage

Figure 2: A cartoon illustrating common challenges associated with information masking: (a) Distribution shift and (b) Information leakage.

### 2.1 COMMON ISSUES IN EVALUATION FRAMEWORKS

To investigate the causal relationship between the identified "salient" features and model decisions, an intuitive approach is to *remove* those features and observe the model predictive power degradation on the altered data. For instance, in image models, researchers often apply a mask to the pixels, replacing them with a fixed value. However, the Remove method raises concerns due to 1) Distribution Shift (Dabkowski & Gal, 2017; Hooker et al., 2018): the masking process introduces artifacts, rendering the modified data out-of-distribution (OOD); and 2) Information Leakage (Rong et al., 2022): the mask can inadvertently reveal class-relevant information, as this information may not be confined to the data value alone. These issues further lead to 3) Ranking Inconsistency (Tomsett et al., 2020; Rong et al., 2022): the explanation evaluation framework may produce unstable rankings depending on the feature masking order (Most Relevant First (MoRF) or Least Relevant First (LeRF)), despite such orders theoretically being irrelevant to the ranking outcome.

Furthermore, frameworks for assessing may lack statistical reliability when applied to diverse datasets or quality metrics (Tomsett et al., 2020; Rong et al., 2022; Brocki & Chung, 2022).

## 2.2 *Fidelity analysis* EVALUATION FRAMEWORKS

**RemOve And Retrain (ROAR)** Hooker et al. (2018) introduced the ROAR evaluation framework to address the issue of distribution shift. In this approach, after features are removed through fixed-value imputation, the model is retrained to adapt to the altered data distribution. The faithfulness of the model is then assessed based on the decline in accuracy of the retrained model.

**RemOve And Debias (ROAD)** Building on ROAR, Rong et al. (2022) identified additional challenges, including information leakage and ranking inconsistency, arising from fixed-value imputation. Using mutual information theory, they proposed the ROAD framework, which employs Noisy Linear Feature Imputation. This method minimizes the revelation of class-relevant information without necessitating retraining, resulting in a consistent ranking of explanation faithfulness.

**Geometric RemOve And Retrain (GOAR)** Park et al. (2023) critically examined the ROAR and ROAD frameworks from a geometric standpoint, highlighting their lack of invariance to coordinate transformations and neglect of directional information in the data's geometric structure. To address these limitations, they proposed the GOAR framework, which incorporates a diffusion model into the ROAR process to purify the modified dataset, offering a coordinate-independent solution.

## 2.3 *Robustness Analysis* EVALUATION FRAMEWORKS

**Adversarial Robustness (AR)** Deep neural networks are known to be vulnerable to adversarial perturbation (Goodfellow et al., 2014), and the EEG-DL models are no exception (Zhang & Wu, 2019). As the common objective of an attack is to maximize the model failure while minimizing the perturbation scale, Hsieh et al. (2020) leveraged this idea and resorted to nuanced adversarial perturbation as an alternative to the brute-force value imputations. Their faithfulness metric is "$Robustness - S$", denoting the maximum perturbation tolerance on the feature subset $S$ that was attributed higher importance by the explanation. Although Hsieh et al. did not directly address the abovementioned issues, we argue that this framework is a capable workaround by exploiting the imperceptible and model parameter-related nature of adversarial perturbation.

**OOD-resistant Adversarial Robustness (OAR)** Expanding on the adversarial robustness (AR) framework, Fang et al. (2024) introduced a novel approach that explicitly accounts for data distribution by incorporating an out-of-distribution (OOD) reweighting block. This block employs a variational graph autoencoder (VGAE) that is trained independently on the unmodified data. The VGAE generates OOD scores for adversarial examples, enabling the reweighting of faithfulness assessments for each explanation method based on its respective OOD score. However, it is important to note that the effectiveness of the VGAE training can be compromised in the presence of significant data variability.

## 2.4 EXPLANATION EVALUATION IN EEG-DL MODEL INTERPRETATION

With the growing understanding of post-hoc explanations in computer vision, there has been a recent expansion into other fields exploring this topic (Turbé et al., 2023; Fang et al., 2024). To the best of our knowledge, there are currently a few peer-reviewed studies that proposed systematic quality evaluations on the subject of post-hoc explanations for DL-based EEG decoders. Apicella et al. (2022) conducted a removal-based study with a 3-layered fully connected network trained for an individual subject from an emotion EEG dataset. In the study, they experimented on MoRF and LeRF removal order from spatial (EEG electrode), spectral (frequency band), and temporal (time sample) perspectives. Cui et al. (2023) also conducted a removal-based study, this time on real EEG datasets and benchmark EEG-DL models. They designed experiments on spatial domain and different scaled temporal domain. Torres et al. (2023) applied trial-level ROAR framework on an autism EEG dataset and its customized CNN. Finally, instead of using removal based method, Sujatha Ravindran & Contreras-Vidal (2023) utilize an EEG generation toolbox with the concept of SNR, which is incompatiable with the idea of domain-specific and removal orders. They designed sensitivity experiments on the synthetic EEG dataset using a toy DL model. However, a fully reproducible explanation evaluation that incorporated the suboptimal evaluation strategies and validated on open EEG datasets and benchmark EEG decoders is still absent, making it difficult to establish trust in previous evaluation outcomes (Singh et al., 2021; Rajpura et al., 2024).

## 3 METHODS

The latest auxiliary model-based frameworks (GOAR, OAR) are underpinned by the auxiliary model's ability to impose distribution constraint on the perturbed data. Preserving distribution for natural multivariate time-series is a non-trivial task, dataset variability for one, the preparatory work can be computationally exhaustive while remaining biased to the limited dataset on which the model was trained (Rong et al. (2022) Appendix B, Fang et al. (2024)). Given these difficulties in justifying such framework design on multivariate EEG, we take the ROAD and AR (SimOAR) frameworks as the cornerstones to develop our explanation faithfulness evaluation framework for EEG, termed multi-domain ROAD (mdROAD) and multi-domain AR (mdAR).

The primary features of EEG are conventionally explored in the spatial (EEG electrode / channel), spectral (frequency band), and temporal (time sample) domain. In this work, we denote the multi-channel EEG sample as $x_{c,t}$ with $N_c$ channels and $N_t$ time points. $S_{c,t}$ is the corresponding saliency map of feature attribution. The imputed EEG data is denoted as $x'_{c,t}$. In spectral domain, we define $X_{c,f} = F_c(x_{c,t})$ as the EEG spectra with $N_f$ frequency bins across channels, where $F_c()$ is the channel-wise fast Fourier transform (FFT). The feature indices in temporal, spatial, and spectral domain to be removed are represented by $\Phi_t$, $\Phi_c$, and $\Phi_f$, respectively. The spatial domain explanation $S_c$ is constructed by $\frac{1}{N_t}\Sigma_c x_{c,t}$, the spectral domain explanation $S_f$ is constructed by $\frac{1}{N_c}\Sigma_f abs(X_{c,f})$, and the temporal domain explanation $S_t$ is constructed by $\frac{1}{N_c}\Sigma_t x_{c,t}$.

### 3.1 MULTI-DOMAIN ROAD FRAMEWORK

In the original ROAD study, Rong et al. (2022) stated that neighboring features are highly correlated, thus a subtle imputation for a pixel can be constructed using the linear interpolation of its neighbors. To ensure the linear relationship to not leak class-related information, a small noise $\epsilon$ is added to the computed interpolation, hence the name "Noisy Linear Imputation". We devised domain-wise feature imputation methods in the spirit of 1) use neighboring features to ensure in-distribution imputation and 2) introduce noise when the imputation is at risk of information leakage.

For spatial domain, the target are *k electrodes* ranked top/last in $S_c$. The imputation result is generated using weighted interpolation of the target's neighboring channels according to the actual electrode montage, the equation can be written as $x'_{c\in\Phi_c,t} = Wx_{c\notin\Phi_c}+\epsilon$, where $W$ stands for a weight matrix for mixing the signals of remaining channels. Since a complete set of neighboring channels includes four direct and four indirect neighbors and the weights should sum up to one, we set $w_d = 1/6$ and $w_{id} = 1/12$ as a Laplacian spatial filter commonly used in channel-wise imputation of EEG data (Banville et al., 2022). In addition, although $\Phi_c$ does not necessary to fall into a connected region, it is often the case that the target electrodes become connected and the solution is solved together as a sparse system.

For spectral domain, the target is a consecutive frequency band that takes *k % of sample power* and is ranked most/least important in $S_f$. The bandwidth is determined exhaustively and differs for each configuration, details are provided in the appendix B.4. In addition to the basic concepts of noisy linear imputation, the spectral feature imputation is also inspired by the "spectrum interpolation" method developed for power line noise removal (Leske & Dalal, 2019). Empirical evidence (He, 2014) suggest that the EEG spectrum has a 1/f-like aperiodic scale-free background component caused by the co-fluctuations of different frequencies (Donoghue et al., 2020), with a power law exponent" falls in [0-3]. To ensure the 1/f-like spectrum structure, the imputation is generated by a real polynomial of degree 3: $P(f) = \Sigma_{i=0}^3 a_i f^{-i}$ fitted onto the sample power spectrum, the imputation equation can be written as $x'_{c,t} = F_c^{-1}(x'_{c,f})$ where $x'_{c,f\in\Phi_f} = P(f)$. Notably, since the amplitude and phase cannot be reconstructed from the power spectrum and there is no knowing how the phase of neighboring frequencies correlate to each other (**?**), only the frequency amplitude is consciously maintained as a means of corruption of the linear relationship.

For temporal domain, the target is a time interval of *k% of series length* whose sum of contribution is ranked top/last by $S_t$, and the temporal feature imputation is emulated using the Multipoint Fractional Brownian Bridge (MFBB) proposed in Friedrich et al. (2020). To briefly touch upon the context, MFBB is a self-similar stochastic series proposed to interpolate sparsely sampled time series, parameterized on a Hurst index $H$, the number of desired timestamps, and more importantly, conditioned on a set of given observations $G_i$ at time instance $t_i$; these properties coincide with our

goal of generating in-distribution imputation from neighboring features. Hurst index is an estimation of the presence of long-range dependency and its degree in a natural time series (Beran et al., 2013; Kannathal et al., 2005), whose value are concluded to reflect certain tendency of value in a time-series. For example, $H < 0.5$ suggests a mean-reverting behavior (Mandelbrot & Van Ness, 1968; Beran et al., 2013). We set $H = 1e - 05$ for the generation of an anti-persistent time series, and the the size of given observation is 3 (placed at the beginning, center and end of target interval) for the effect of minimum class information preservation.

Informally, the MFBB is applying constraint on a stochastic process $B(t)$ (Fractional Brownian Motion (FBM) (Dieker, 2004)), and by definition the imputation function can be written as equation 1. $B(t)$ is characterized by covariance $\langle B(t_1), B(t_2) \rangle = 1/2(|t_1|^{2H} + |t_2|^{2H} - |t_1 - t_2|^{2H})$ and implemented using the method proposed by Davies & Harte (1987). $t$ is a time instance of the imputation target, and $t_i, t_j$ refers to the time instance of previous and next given observation. $\sigma_{ij}$ is the derived autocovariance of $S$. The derivation of the imputation function and more context behind the definitions are provided in the appendix.

$$x'_{c,t \in \Phi_t} = B(t) - [B(t_i) - G_i]\sigma_{ij}^{-1}\langle B(t), B(t_j) \rangle \tag{1}$$

## 3.2 Multi-domain AR framework

The faithfulness measurement in the original AR framework is the minimum perturbation magnitude required to successfully degrade model performance on a designated feature subset. However, empirically we found an impartial measurement of perturbation tolerance on EEG is infeasible providing the fact that our experiment incorporated a wide variety of configurations (experiment subjects, dataset properties, model structures, feature domains), which is in line with the literature (Zhang & Wu, 2019; Meng et al., 2023).

Considering our objective of domain specific information masking, we designed domain-wise imputations with adversarial example instead of conducting domain-specific attacks, which is beyond the scope of this study. Our method is conceptually similar to CutMix augmentation (Yun et al., 2019) in which a region from one sample is removed and patched using another sample. In our case, the target features will be imputed with the corresponding features from its adversarial example $x^{Adv}$. Theoretically, this procedure should imperceptibly move the class-relevant features toward the irrelevant direction along the explanation-based region and degrade the model performance (Fawzi et al., 2017). The adversarial examples in this study were generated with untargeted Projected Gradient Descent (PGD) attack, which is a multi-step first-order attack method proven to be effective in several scenarios (Madry et al., 2017; Meng et al., 2023). The PGD formula and parameter setting are presented in the appendix. For mdAR framework, the imputation target selection is identical to mdROAD framework, and the imputation functions for spatial, spectral and temporal domain can be written as $x'_{c \in \Phi_c} \leftarrow x^{Adv}_{c \in \Phi_c}$, $X'_{f \in \Phi_f} \leftarrow X^{Adv}_{f \in \Phi_f}$ and $x'_{t \in \Phi_t} \leftarrow x^{Adv}_{t \in \Phi_t}$, respectively.

# 4 Experimental Setup

## 4.1 EEG Datasets and Decoding Neural Networks

EEG datasets harbor distinctive task-related characteristics, and differently structured decoders have strength in certain feature domains. To support the generalizability of the proposed framework, we cooperated three well-studied public EEG datasets and three lightweight CNN-based models for this study.

**Open Multivariate EEG datasets** We selected one time-asynchronous and two time-synchronous multivariate EEG datasets, referred to as sensory motor rhythm (SMR) dataset, event-related negativity (ERN) dataset and steady state visual evoked potential (SSVEP) dataset according to the BCI paradigm they represented. The time-asynchronous SMR dataset comes from BCI Competition Dataset 2A Brunner et al. (2008), containing EEG desynchronization of imaged movements in sensorimotor cortex. ERN dataset comes from "BCI-challenge" on Kaggle (Jérémie Mattout & Kan, 2014), reflects time-locked EEG amplitude change elicited by oddball events. SSVEP datasets comes from MAMEM SSVEP experiment 2 (Martinez et al., 2007), the EEG recording around occipital region synchronizes with given visual stumli at specific frequencies. Detailed dataset information and the dataset preprocessing procedures are described in the appendices.

**EEG decoding neural networks** Convolutional Neural Networks (CNN) operation mimics conventional spatial or temporal filtering in EEG feature extraction, hence CNN-based model structures are often adopted for EEG decoding. EEGNet (Lawhern et al., 2018) is a compact model with a temporal convolution layer, a depthwise convolution layer, and a separable convolution layer. The model has been tested on abundant EEG research. SCCNet (Wei et al., 2019) was proposed for the SMR dataset, later utilized to analyze different datasets (Pan et al., 2022). It features a spatial convolution layer followed by a spatio-temporal convolution layer. InterpretableCNN (Cui et al., 2022) was proposed for EEG drowsiness recognition, it consisted of a pointwise convolution layer and a depthwise convolution layer. Unlike the common EEG-DL model structure, its batch normalization layer tracks batch moments rather than running moments. To alleviate the influence of individual EEG variability, all of our models are trained in a subject specific manner, the implementation and training settings are provided in the appendices. Average accuracies of the three models are $\{72.80\%, 70.86\%, 72.21\%\}$ on the SMR dataset, and $\{72.36\%, 60.59\%, 72.23\%\}$ on the SSVEP dataset; the roc-auc score is $\{87.81\%, 87.72\%, 88.77\%\}$ on the ERN dataset.

## 4.2 FEATURE ATTRIBUTION METHODS

The vast literature has concluded their takes on the quality of different explanation methods, however, the results are diverse as a product of varying data characteristics and evaluation strategies. Regardless of previous faithfulness evaluation results, we selected 6 common back-propagation based feature attribution methods. Concerning whether the sign of explanation holds class-relevant information, a crude consensus is that it depends on the underlying data characteristics (Bach et al., 2015; Smilkov et al., 2017; Ancona et al., 2017). Since this matter was never discussed in the context of EEG, we decided to investigate signed explanations and their absolute values as two different explanations. Altogether, 6+4 methods will be evaluated along with a *Random* baseline, namely *Gradient* w/wo absolute (GD/GDA) (Simonyan et al., 2013), *Gradient×Input* w/wo absolute (GI/GIA) (Shrikumar et al., 2017), *Smoothgrad* w/wo absolute (SG/SGA) (Smilkov et al., 2017), *Smoothgrad Squared* (SG) (Hooker et al., 2018), *Vargrad* (VG) (Adebayo et al., 2018), and *Integrated Gradient* with canonical baseline 0 w/wo absolute (IG/IGA) (Sundararajan et al., 2017). The implementation details are provided in the appendices.

## 4.3 EXPLANATION FAITHFULNESS MEASUREMENTS

The faithfulness metric within the notion of deficiency between performance degradation and feature perturbation should be able to capture it through different levels of explanation-based feature masking. Instinctively, the model performance of MoRF order should have a sharp decrease right after masking begins, and the decrease in LeRF performance should be modest (Hooker et al., 2018). According to such expectation of accuracy-ratio curve behavior, mainstream qualitative metrics are area-centric as equation 2 (Tomsett et al., 2020; Apicella et al., 2022; Brocki & Chung, 2022; Cui et al., 2023). Each post-hoc explanation method will correspond to two performance curves of different masking order in one framework-dataset-model-domain configuration. Taking into account one or both strategies, we proposed three normalized area metrics: Area Over Curve (AOC), Area Between Curve (ABC), and Area Under Curve (AUC).

$$AOC = {}^1\!/_{K+1} \sum\nolimits_{k=0}^{K} \left[ \frac{Acc(x^0) - Acc(x_{\mathrm{M}}^k)}{Acc(x^0) - {}^1\!/_{|Class|}} \right]$$

$$ABC = {}^1\!/_{K+1} \sum\nolimits_{k=0}^{K} \left[ \frac{Acc(x_{\mathrm{L}}^k) - Acc(x_{\mathrm{M}}^k)}{Acc(x^0) - {}^1\!/_{|Class|}} \right] \qquad (2)$$

$$AUC = {}^1\!/_{K+1} \sum\nolimits_{k=0}^{K} \left[ \frac{Acc(x_{\mathrm{L}}^k) - {}^1\!/_{|Class|}}{Acc(x^0) - {}^1\!/_{|Class|}} \right],$$

where $x_{\mathrm{M}}^k$ ($x_{\mathrm{L}}^k$) denotes the input with most (least) important $k$ percent features masked, K is the maximum masking ratio. $Acc(x)$ stands for the classification power of model on input $x$. Higher measurements indicate that the curves' behavior aligns more closely with expectations, reflecting greater faithfulness. An illustrated example of the metrics is displayed in Figure 1 b.

### 4.4 Framework Consistency on Masking Order

We use Spearman's ranking correlation coefficient to quantify the consistency of framework results in different masking orders. We first rank the explanation methods based on their impact on model performance at varying masking ratios, where a greater decrease in performance results in a higher ranking for Most Relevant Features (MoRF) and a lower ranking for Least Relevant Features (LeRF), with ranks ranging from 1 (best) to 11 (worst). Next, we calculate the ranking correlation coefficients for each ratio up to 50%. The average correlation coefficient across these ratios reflects the framework's consistency across various combinations of domains, datasets, and models. The Spearman's ranking correlation coefficient $\rho$ measures the monotonic correlation between two ranks, the possible range is [-1,1], with positive (negative) results suggest the degree of similarity (dissimilarity). For two ranks $R_M$, $R_L$, the correlation coefficient $\rho_{R_M,R_L}$ is defined as 3. $cov, \sigma$ denotes the covariance and standard deviation of the rankings, respectively.

$$\rho_{R_M,R_L} = \frac{cov(R_M, R_L)}{\sigma_{R_M}, \sigma_{R_L}}. \tag{3}$$

## 5 Experiments

In this section, we first report the evaluation results using our proposed frameworks, before validating the frameworks quantitatively and qualitatively. We also address the possible issue of using unsigned explanation methods for data in signal format with visualized examples.

Table 2: Comparison of faithfulness scores for various feature attribution methods, averaged across different dataset-model configurations. Higher values indicate greater faithfulness. Highlighted cells represent the "most faithful" method, and the superscripts mark the top-3 highest faithfulness measurement within each column.

| method | Spatial | | | | | | Temporal | | | | | | Spectral | | | | | |
|---|---|---|---|---|---|---|---|---|---|---|---|---|---|---|---|---|---|---|
| | mdROAD | | | mdAR | | | mdROAD | | | mdAR | | | mdROAD | | | mdAR | | |
| | AOC | ABC | AUC | AOC | ABC | AUC | AOC | ABC | AUC | AOC | ABC | AUC | AOC | ABC | AUC | AOC | ABC | AUC |
| GD | .491±.181 | .024±.042 | .463±.095 | .492±.143 | .002±.002 | .508±.143 | .405±.135 | .014±.015 | .598±.139 | .498±.133 | .007±.011 | .500±.131 | .523±.115[1] | .232±.039[2] | .709±.125[2] | .602±.140[1] | .258±.076[1] | .656±.141[1] |
| GI | .590±.115[2] | .152±.097[2] | .557±.194 | .494±.151 | .008±.010 | .513±.149 | .439±.123 | .077±.047 | .632±.144 | .511±.128 | .034±.033 | .519±.129 | .474±.112 | .175±.072 | .700±.129 | .523±.114[3] | .155±.106[3] | .620±.165[3] |
| SG | .489±.182 | .026±.046 | .465±.095 | .492±.143 | .002±.002 | .508±.143 | .405±.134 | .013±.015 | .598±.138 | .498±.133 | .007±.011 | .500±.130 | .523±.116[2] | .234±.038[1] | .711±.124[1] | .600±.139[2] | .250±.079[2] | .651±.147[2] |
| SS | .544±.192 | .148±.065 | .604±.156[1] | .542±.143[1] | .138±.036[2] | .596±.145[1] | .445±.134[3] | .103±.045[3] | .658±.135[2] | .529±.120[1] | .098±.055 | .569±.149[1] | .505±.118 | .208±.066 | .702±.129 | .515±.105 | .152±.113 | .626±.165 |
| VG | .543±.193 | .145±.067 | .602±.148[2] | .536±.147 | .126±.033 | .590±.142[2] | .443±.135 | .099±.044 | .656±.133 | .526±.121[2] | .093±.053[3] | .567±.149[2] | .511±.128[3] | .186±.085 | .674±.143 | .501±.106 | .115±.113 | .578±.179 |
| IG | .596±.115[1] | .166±.093[1] | .563±.194 | .494±.151 | .010±.012 | .514±.149 | .450±.128[1] | .095±.044 | .645±.136 | .516±.127 | .045±.035 | .529±.131 | .475±.114 | .176±.073 | .701±.128 | .521±.113 | .154±.106 | .620±.166 |
| GDA | .542±.192 | .142±.069 | .600±.158 | .541±.142[2] | .137±.037[3] | .596±.146[3] | .445±.133[2] | .102±.045 | .657±.136[3] | .510±.129 | .048±.046 | .534±.137 | .506±.118 | .203±.063[3] | .698±.133 | .514±.104 | .145±.117 | .615±.171 |
| GIA | .552±.177[3] | .154±.055[2] | .602±.166 | .533±.150 | .122±.036 | .589±.141 | .444±.133 | .103±.047[2] | .659±.136[1] | .519±.124 | .063±.044[2] | .545±.135 | .499±.116 | .200±.065 | .701±.129 | .514±.100 | .128±.118 | .597±.178 |
| SGA | .543±.192 | .143±.066 | .600±.159 | .541±.142[2] | .137±.037[1] | .596±.145[1] | .445±.134[3] | .102±.045 | .657±.135 | .510±.130 | .048±.046 | .535±.136 | .506±.117 | .206±.062 | .700±.132 | .516±.105 | .147±.116 | .616±.173 |
| IGA | .552±.179 | .154±.049 | .602±.164[3] | .532±.150 | .120±.036 | .587±.142 | .443±.134 | .100±.047 | .657±.137 | .519±.123[3] | .068±.044[1] | .549±.137[3] | .501±.116 | .203±.063 | .702±.128[3] | .517±.101 | .133±.115 | .601±.177 |
| RD | .483±.170 | .005±.005 | .518±.171 | .474±.146 | .001±.001 | .526±.146 | .394±.134 | .006±.005 | .599±.135 | .487±.134 | .003±.004 | .504±.132 | .400±.130 | .002±.002 | .519±.114 | .478±.136 | .003±.008 | .476±.097 |

### 5.1 Faithfulness Evaluation results

The faithfulness measurements of the 6+4 explanation methods are presented in table 2. The faithfulness of *Random* baseline explanation is trivially the worst, aligning with the assumption that it's supposed to carry no class-related information. Additionally, methods with absolute are generally measured to be more faithful in spatial and temporal domain than their without absolute counterparts. Considering EEG data characteristics where relative changes are often more meaningful, this result is not at all surprising. As for the with/without absolute comparison in spectral domain, the without absolute methods turn out to achieve superior faithfulness measurements than with absolute methods, we address this phenomenon in section 5.3.

### 5.2 Quantitative Framework Validation with Masking Order Consistency

Table 3 shows Spearman's ranking correlation coefficient between masking orders of each framework-domain-dataset-model configuration. Throughout dataset and model configurations, the mdAR framework seems to produce more consistent result between masking order when comparing to the mdROAD framework. At the end of the day, the feature interpolation in the mdROAD are handcrafted with human knowledge on the EEG data, and we believe such phenomenon is a reflection of stronger bias enforced in mdROAD framework than in mdAR framework. The complete ranking of each framework-domain-dataset-model is provided in the appendix D.2.

Table 3: Comparison of the two imputation techniques: Multi-Domain Remove and Debias (mdROAD) and Multi-Domain Adversarial Robustness (mdAR) regarding framework reliability for faithfulness evaluation. Reliability is evaluated based on the consistency of rankings for feature attribution methods derived from the Most Relevant Features (MoRF) and Least Relevant Features (LeRF) strategies, using Spearman's $\rho$ for each framework-domain-dataset-model configuration. Bold text highlights configurations where there is greater consistency in masking order between the two frameworks.

| Framework | Domain | SMR | | | ERN | | | SSVEP | | |
|---|---|---|---|---|---|---|---|---|---|---|
| | | EEGNet | ICNN | SCCNet | EEGNet | ICNN | SCCNet | EEGNet | ICNN | SCCNet |
| | Spatial | .136±.361 | .377±.280 | .557±.197 | **.854±.098** | .583±.190 | .563±.203 | .653±.199 | .733±.095 | .707±.179 |
| mdROAD | Temporal | .548±.162 | .174±.267 | **.785±.149** | .538±.162 | **.700±.178** | .728±.164 | **.570±.227** | **.764±.141** | **.353±.259** |
| | Spectral | .788±.120 | .101±.318 | **.362±.219** | .383±.249 | .115±.244 | .166±.381 | .318±.340 | .077±.324 | .023±.263 |
| | Spatial | **.830±.066** | **.816±.051** | **.821±.059** | .767±.105 | **.773±.099** | **.833±.078** | **.881±.071** | **.870±.052** | **.862±.108** |
| mdAR | Temporal | **.921±.079** | **.790±.193** | .638±.130 | **.904±.096** | .614±.254 | **.805±.062** | .434±.281 | .715±.071 | -.040±0.370 |
| | Spectral | **.813±.117** | **.755±.091** | .330±.218 | **.681±.226** | **.482±.207** | **.625±.113** | **.755±.082** | **.592±.091** | **.811±.061** |

## 5.3 FREQUENCY DISTORTION IN UNSIGNED MODEL EXPLANATION

In explanation faithfulness measurement results, we notice that the relative rank of methods w/wo absolute in spectral domain are inconsistent with the other two domains, hereby we address this abnormality with a visualized example. As fig. 3 (a) shows, the period of the absolute explanation is half of the original signal, in other words, the frequency is doubled, and the power of the corresponding double will increase. The signal-like explanations can be visualized in a frequency by time representation by applying short time Fourier transform, and Fig. 3 (b) shows the transformed explanation before and after absolute, within which the "significant frequency" clearly shifted to different multiples of the stimulus frequency. The reason that the shifted band does not manifest at a perfect double is because of the harmonic frequencies invoked by the visual stimuli and the interference of other brain activities.

To support this reasoning of inconsistent spectral domain faithfulness ranking, we conduct an "frequency correction" experiment on SSVEP dataset based on the mdAR framework. In the experiment, the target $\Phi_f$ now are frequencies whose *amplitude* were ranked top/last k% in the $S_f$, but the imputation function becomes $X'_{f/2} \leftarrow X^{Adv}_{f/2}$ for $f \in \Phi_f$. We observed improvements in faithfulness metrics that take MoRF order into account, the difference of {AOC, ABC, AUC} on GDA is {+.062, +.045, -.015}, on GIA is {+.067, +.031, -.036}, on SGA is {+.062, +.049, -.013}, and on IGA is {+.063, +.023, -.030}. However, due to the frequencies being mixed non-linearly in EEG, the perfect frequency correction for unsigned model explanation requires extra effort beyond the scope of this study. Nevertheless, the aforementioned frequency distortion phenomenon is something EEG-DL researchers should be extra careful when adopting XAI methods that embody some sign-elimination when interpreting spectral features.

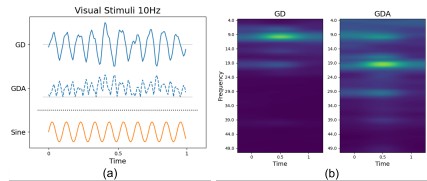

Figure 3: Frequency distortion observed in unsigned explanations. (a) Temporal saliency of Gradient (GD) and Gradient with Absolute (GDA) comparing to a reference 10-Hz sinusoidal signal. (b) Shows a comparison of the spectrograms for the temporal saliency of GD and GDA, highlighting that GDA exhibits a 20-Hz component attributed to the incorporation of absolute saliency.

## 5.4 QUALITATIVE FRAMEWORK VALIDATION WITH NEUROSCIENTIFIC EVIDENCE

We complement the quantitative validation of evaluation frameworks by visualizing the explanations evaluated as the most and least faithful, and interpret them with neuroscientific knowledge. We select EEGNet from one repeat for example and the subjects it achieved the best classification performance, which are subject 3 from SMR dataset with 0.8923 accuracy, subject 22 from ERN dataset with 0.9828 roc-auc score, and subject 11 from SSVEP dataset with 0.92 accuracy. The faithfulness evaluation values are provided in the figure.

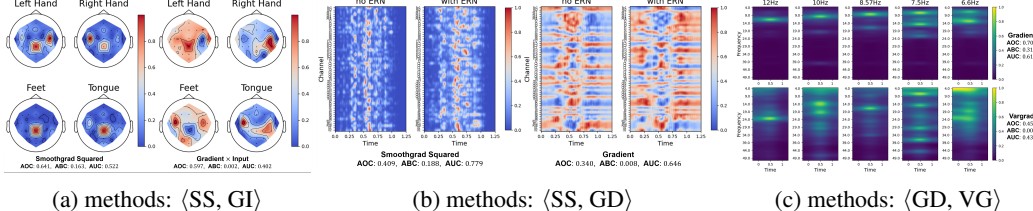

(a) methods: ⟨SS, GI⟩          (b) methods: ⟨SS, GD⟩          (c) methods: ⟨GD, VG⟩

Figure 4: ⟨Most, Least⟩ faithful explanation method visualization, saliency values normalized to 0-1, EEGNet for example, faithfulness measurement from mdAR framework. (a) Spatial domain explanation on SMR dataset. (b) Temporal domain explanation on ERN dataset. (c) Spectral domain explanation on SSVEP dataset.

In the Spatial domain with SMR case, we can see *Smoothgrad Squared* better captured motor cortex activations, especially the contralateral pattern in class "Left Hand" and "Right Hand", and response close to the longitudinal fissure of class "Feet".In the temporal domain with ERN case, *Smoothgrad Squared* shows a converged activation around 500 milliseconds after cue onset, while the magnitude in *Gradient×Input* are rather scattered. In the spectral domain with SSVEP case, the significant frequency responses are duly intensified at the stimuli and their harmonic frequencies in *Gradient*; in *Vargrad* the responses are discernable but diluted. As models with better convergence are presumed to extract data characteristics well, we found that explanations evaluated as more faithful by our framework did contain patterns aligned with neuroscientific knowledge (Pfurtscheller et al., 2006; Hajcak et al., 2005; Martinez et al., 2007).

# 6 CONCLUSION

We introduce a novel adversarial information masking (AIM) approach to enhance in-distribution information masking, addressing key gaps in the assessment of faithfulness for post-hoc explanations in deep learning. To validate the AIM method, we conduct the first quantitative comparison of faithfulness assessment frameworks across various architectures, datasets, and domains. Through these efforts, we successfully identify effectiveness of post-hoc explanation methods in EEG-DL, thus furthering our understanding of model behavior and improving their explainability. Future research could focus on refining these frameworks and exploring their applicability to a wider range of multivariate time series and sequential data contexts.

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

## A TEMPORAL FEATURE IMPUTATION IN MDROAD

### A.1 FRACTIONAL BROWNIAN MOTION

**General Representations** Fractional Brownian motion (FBM) is a self-similar stochastic process designed to practically model natural time-series with minimum mathematical difficulty. In the original definition, the FBM is modeled with a Gaussian process and formulated as a Riemann-Liouville integral (Loeve, 1948) of . Later work by Mandelbrot & Van Ness (1968) introduced FBM represented in Weyl integral, which has stationary increments and a simpler covariance function. Denoting $X_H(t)$ as the observation of the FBM at $t = t$, and $X(t)$ is real noise (ordinary Brownian motion modeled by Gaussian white noise), for $t > 0$, FBM is written as $X_H(t) - X_H(0) = \frac{1}{\Gamma(H+\frac{1}{2})}\{\int_{-\infty}^{t}(t-s)^{H-\frac{1}{2}}dX(s) - \int_{-\infty}^{0}(0-s)^{H-\frac{1}{2}}dX(s)\}$. The self-similar property can be represented as $\Delta(X_H(t+\tau), X_H(t)) = h^{-H}\Delta(X_H(t+\tau), X_H(t))$. The covariance function of FBM in Weyl intergral representation can be written as $\langle X_H(t_0)X_H(t_1)\rangle = \frac{\Gamma(1-2H)cos(H\pi)}{2H\pi}(|t_0|^{2H} + |t_1|^{2H} - |t_0 - t_1|^{2H})$; for a fixed $H$, the scalar term is also fixed. The full derivation of covariance function can be found in item $\langle 5.1 \rangle$ of Mandelbrot & Van Ness (1968).

Although EEG is a non-stationary, non-linear and noisy signal (Klonowski, 2009), the imputed signal is short and the mdROAD framework emphasize on utilizing the distribution of neighbors, we view the time-series in the masked short interval as quasi-stationary and adopted the multipoint fractional Brownian bridge (MFBB) method. Previous MFBB utilization on EEG appeared in Ma et al. (2023) .

**Davies-Harte method for Fractional Brownian Motion Simulation** Several algorithms are developed to simulate FBM, some simulated results have exact property of FBM while some algorithm choose to approach the properties by approximation, considering benefits such as computation speed. We used method from Davies & Harte (1987) to simulate exact FBM for efficient series generation.

Given a one dimensional fractional Gaussian noise of length $n$: $X = (X(0), X(1)...X(n-1))^T$, its covariance function denoted as $\gamma(\cdot)$. In our case $\gamma(k) = \frac{1}{2}(|k+n|^{2H} + |k-n|^{2H} - |2k|^{2H})$, $n, k = 0, 1, 2....$. $\Gamma$ is a Toeplitz matrix constructed as 4:

$$
\Gamma_{n \times n} = \begin{pmatrix} \gamma(0) & \gamma(1) & \cdots & \gamma(n-1) \\ \gamma(1) & \gamma(0) & \cdots & \gamma(n-2) \\ \vdots & \vdots & \ddots & \vdots \\ \gamma(n-1) & \gamma(n-2) & \cdots & \gamma(0) \end{pmatrix} \tag{4}
$$

The main idea is to find the square root $G$ of $\Gamma$ in the sense that $\Gamma = GG^T$. Embed $\Gamma$ in the upper left corner of circulant covariance matrix $C$ constructed using similar process with size $M \geq 2N - 1$, written as 5, within which $c_j = \gamma(j)$ for $0 \leq j \leq \frac{M}{2}$ and $c_j = \gamma(j)$ for $\frac{M}{2} \leq j \leq M - 1$.

$$
C_{M \times M} = \begin{pmatrix} c_0 & c_1 & c_2 & \cdots & c_{m-1} \\ c_{m-1} & c_0 & c_1 & \cdots & c_{m-2} \\ c_{m-2} & c_{m-1} & c_0 & \cdots & c_{m-3} \\ \vdots & \vdots & \vdots & \ddots & \vdots \\ c_1 & c_2 & c_3 & \cdots & c_0 \end{pmatrix} \tag{5}
$$

Circulant matrix $C$ has the representation $Q\Lambda\bar{Q}^T$, $\bar{Q}$ is the complex conjugate of $Q$. $\Lambda$ is the diagonal matrix of eigenvalues of $C$ that $\Lambda = diag(\lambda_0, \lambda_1, ...\lambda_{M-1})$, and $Q$ is unitary matrix that $Q\bar{Q}^T = 1$. The entries of Q is defined in 6 and the eigenvalues are given by discrete fourier transform of the first row in $C$ 7:

$$
q_{jk} = \frac{1}{\sqrt{M}} \ exp(-2\pi i \frac{jk}{M}) \quad \text{for } k = 0, 1, 2....M - 1, \tag{6}
$$

$$
\lambda_k = \Sigma_{j=0}^{M-1} \ c_j \ exp(2\pi i \frac{jk}{M}) \quad \text{for } k = 0, 1, 2....M - 1, \tag{7}
$$

Let $S = Q\Lambda^{\frac{1}{2}}\bar{Q}^T$ so that $C = S\bar{S}^T$, a standard normal complex sequence $Z$ multiplied by $S$ will satisfy the desired property; that is, the first $N$ terms in $SZ$ is the simulated FBM.

For a standard normal variable $v$, multiply by $\sqrt{M}Q$ is as if taking discrete fourier transform (DFT) of $v$, and multiply by $\frac{1}{\sqrt{M}}\bar{Q}^T$ is as if taking inverse discrete fourier transform (iDFT) of $v$ 8. Consequently, $SZ = iDFT(\Lambda^{\frac{1}{2}}DFT(Z))$. (Brockwell & Davis, 1991; Wood & Chan, 1994; Dieker, 2004; Banna et al., 2019)

$$
\sqrt{M}Qv = (\Sigma_{k=0}^{M-1} \ v_k \ exp(-2\pi i \frac{jk}{N}))_{k=0}^{M-1}
$$
$$
\frac{1}{\sqrt{M}}\bar{Q}^T v = (\frac{1}{M}\Sigma_{k=0}^{M-1} \ v_k \ exp(2\pi i \frac{jk}{N}))_{k=0}^{M-1} \tag{8}
$$

## A.2 MULTIPOINT FRACTIONAL BROWNIAN BRIDGE

A fractional Brownian bridge (FBB) is defined as a FBM starting from 0 at $t = 0$ and ends at $X_T$ when $t = T$. FBB is constructed with a Gaussian process conditioned on $X_T$, its one- and two-point correlation function are:

$$
\langle X(t_1) \rangle = \frac{\langle X(t_1)\delta(X(T) - X_T) \rangle}{\langle \delta(X(T) - X_T) \rangle}
$$
$$
\langle X(t_1)X(t_2) \rangle = \frac{\langle X(t_1)X(t_2)\delta(X(T) - X_T) \rangle}{\langle \delta(X(T) - X_T) \rangle} \tag{9}
$$

A function satisfy equation 9 can be constructed as $X_{FBB}(t) = X(t) - (X(T) - X_T)\frac{\langle X(t)X(T) \rangle}{\langle X^2(T) \rangle}$ to generate FBB. A complete derivation can be found in the appendix of Delorme & Wiese (2016).

Multipoint fractional Brownian bridge is the ordinary FBB generalized to an arbituary number of prescribed points. Considering the MFBB is conditioned on a set of points $X_i$ at $t_i$ for $i = 1, 2....n$, the one- and two-point conditional moments are:

$$\langle X(t_1)|\{X_i, t_i\}\rangle = \frac{\langle X(t_1)\Pi_{i=1}^n \delta(X(t_i) - X_T)\rangle}{\Pi_{i=1}^n \langle \delta(X(t_i) - X_T)\rangle}$$

$$\langle X(t_1)X(t_2)|\{X_i, t_i\}\rangle = \frac{\langle X(t_1)X(t_2)\Pi_{i=1}^n \delta(X(t_i) - X_T)\rangle}{\Pi_{i=1}^n \langle \delta(X(t_i) - X_T)\rangle} \quad (10)$$

Using similar process as FBB, a function can be constructed to satisfy 10 as $X_{MFBB}(t) = X(t) - (X(t_i) - X_i)\frac{\langle X(t)X(t_j)\rangle}{\langle X(t_i)X(t_j)\rangle}$ for $i, j = 1, 2....n$. A complete derivation can be found in the appendix of Friedrich et al. (2020).

## B   IMPLEMENTATION DETAILS

### B.1   EXPLANATION METHODS

All explanations are generated with python Captum (Kokhlikyan et al., 2020) package, all sign are kept and the explanations were not normalized until visualization. The absolute values and masking were conducted within single trial. When visualizing explanations, we compute min-max normalization to scale the values to 0-1 after averaging across unwanted dimensions.

**Gradient and Gradient $\times$ Input**   Gradient $E_{GD}(x)$ (Simonyan et al., 2013) are the gradient of class score with regard to input. Gradient $\times$ Input $E_{GI}(x)$ (Ancona et al., 2017) is obtained through element-wise multiplying Gradient with original input.

$$E_{GD}(x) = \frac{\partial S_c(x)}{\partial x}$$
$$E_{GI}(x) = E_{GD}(x) \odot x \quad (11)$$

**Smoothgrad , Smoothgrad Squared and Vargrad**   Smoothgrad $E_{SG}(x)$ , Smoothgrad Squared $E_{SS}(x)$ and Vargrad $E_{VG}(x)$ (Smilkov et al., 2017; Adebayo et al., 2018) are ensemble explanation methods that can reduce visually noisy explanation maps. Here we took Gradient as the primitive explanation method, the number of random samples $N$ for ensemble explanation methods are set to be 16, and their noise level $\epsilon$ set as $\sim N(0, 1e - 2)$.

$$E_{SG}(x) = \frac{1}{N}\sum_{i=1}^N (E_{GD}(x + \epsilon))$$
$$E_{SS}(x) = (\frac{1}{N}\sum_{i=1}^N (E_{GD}(x + \epsilon)))^2$$
$$E_{VG}(x) = Variance(E_{GD}(x + \epsilon)) \quad (12)$$

**Integrated Gradient**   Integrated Gradient $E_{IG}(x)$ (Sundararajan et al., 2017) sums over the values from "baseline" $\bar{x}$ along a interpolation path up to the actual Gradient. Although the "baseline" has been proven to have nonnegligible influence on the explanation result, we follow typical setting to set the baseline as zero, and the default scaling variable $\alpha = 50$.

$$E_{IG}(x) = (x - \bar{x}) \times \int_0^1 \frac{\partial S_c(\bar{x} - \alpha(x - \bar{x}))}{\partial x} d\alpha \quad (13)$$

### B.2   DATASET AND PREPROCESSING

**Sensory Motor Rhythm (SMR)**   Motor imagery (MI) reflects endogenous activity in the sensorimotor cortex induced by imagined movement. The SMR dataset in this study comes from BCI Competition Dataset 2A (Brunner et al., 2008), it consisted of 22 channel EEG data recorded at 250Hz sampling rate from 9 subjects performing 4 MI tasks (left hand, right hand, feet and tongue).

The dataset contains one training session and one evaluation session for each subject recorded on different dates. A session includes 72 trials for each of the 4 tasks. The preprocessing followed Wei et al. (2019) by downsample to 125 Hz and epoch to [-0.5, 4] seconds post cue onset.

**Feedback Error-Related Negativity (ERN)**   Feedback Error-Related Negativity (ERN) is a time-locked amplitude component that can be observed after the subject encounters an erroneous event. The dataset comes from (Jérémie Mattout & Kan, 2014) and is available as the early stage release in the "BCI challenge" on Kaggle. The dataset consisted of 56 channel EEG data recorded at a 600Hz sampling rate from a total of 16 subjects executing P300 speller task. The experiment had a total of 340 trials from 5 sessions, where we split the first 300 trials as the training set and the remainder as the test set. The preprocessing followed Pan et al. (2022) by downsample to 128 Hz, bandpass filter to 1-40 Hz and epoch to [0, 1.25] seconds post cue onset.

**Steady state visual evoked potential (SSVEP)**   Steady state visual evoked potential (SSVEP) are quasi-periodic oscillatory responses that occur in the occipital cortex when a person is visually stimulated by flickering of a specific frequency (Wang et al., 2016). The dataset in this study comes from MAMEM SSVEP experiment 2 (Martinez et al., 2007), it consisted of 256 channel EEG data recorded at 250Hz sampling rate from 11 subjects, with stimuli in 5 frequencies (6.66, 7.50, 8.57, 10.00, and 12.00 Hz). The experiment has 5 sessions, we used the first 4 sessions as our training set and the last as the test set. The preprocessing followed Pan et al. (2022) by downsample to 125Hz, bandpass filter to 1-50 Hz, and epoch the original trials into 1 second segments.

## B.3   MODELS AND TRAINING SETTING

The models and training procedure were implemented using pytorch framework (Paszke et al., 2019). We repeatedly trained 5 set of models using different random seeds, and the quantitative results in the manuscript were averaged across the repeats, except for the distorted frequency correction experiment which was conducted on one set of the models. In the total training of 500 epochs, the model with best test accuracy will be taken to generate explanation and conduct masking experiment. The learning rate were initially set as 5e-4 with a 0.01 decay every ten epochs using ExponentialLR. Adam optimizer were used. Batch size was set as 32 for SMR dataset, 32 for ERN dataset and 25 for SSVEP dataset. As the models were trained for individual subjects, we assume that the variance between data samples or batches are ignorable; thus, the batch normalization layers were left unmodified.

The parameter of EEGNet and InterpretableCNN trained on SMR and ERN dataset followed the default setting that EEGNet: {F1=8, F2=16, D=2} and InterpretableCNN: {N1=16, d=2}, for SSVEP we used EEGNet: {F1=100, F2=10, D=8} and InterpretableCNN: {N1=100, d=8}.

## B.4   DETAILS FOR IMPUTATION

**Spectral domain target search**   For spectral domain target frequency band $B$, firstly we assume for each center frequency $b_c$ we can find frequency band $B_L$ ($b_{left}$–$b_c$ Hz) and $B_R$ ($b_c$–$b_{right}$ Hz) whose power are both $\frac{k}{2}$% of sample power, $b_{left}$ and $b_{right}$ can be exhaustively determined by gradually adding up the power from each side of $b_c$. Since the datasets were bandpass filtered in the preprocessing procedure, special cases occur when $b_{left}$ or $b_{right}$ hits the band limit before power under the $B_L$ or $B_R$ meet $\frac{k}{2}$% of sample power, our solution is to continue the search in the opposite direction.

**Imputation ratio setting**   Considering artifact introduced, only the performance up to 50% masking ratio will be taken into quantitative analysis. For spatial domain, the masking ratio range from 1 to the half number of channels. For computation time concerns, the interval of spectral and temporal domain are set to 5%.

**Attack method for mdAR framework**   Project gradient descent (PGD) attack starts by adding a small noise $\xi$ to benign data, then iteratively take small gradient steps of size $\alpha$ as an optimization of 1) minimize attack magnitude and 2) maximize effect of attack. To constraint the result to fall in

a $\varepsilon - \ell_2$ or $\varepsilon - \ell_\infty$ neighborhood, the result is projected back onto the neighborhood after each step. The expression of PGD can be written as 14.

$$x_0^{Adv} = x + \varepsilon$$
$$x_{iter}^{Adv} = Proj_\varepsilon(x_{iter-1}^{Adv} + \alpha \times sign\nabla Loss(x_{iter}^{Adv}, y))$$
(14)

Our experiment conducted untargeted attack ($y = y_{true}$), and used cross-entropy to be the $Loss$ function. The neighborhood is a $\ell_2$ ball with radius equal to the extreme values of original data. By empirically testing for effective attack for all datasets and models, we set $\alpha$ to be 2 and iterations to be 10.

## C    EXTENDED FIGURE OF PERFORMANCE-RATIO CURVES

The extended figures of performance-masking ratio curves of each dataset is provided in figure 5, 6 and 7. The curves presented are results firstly averaged across dataset subjects, and then averaged across 5 differently random seeded set of models.

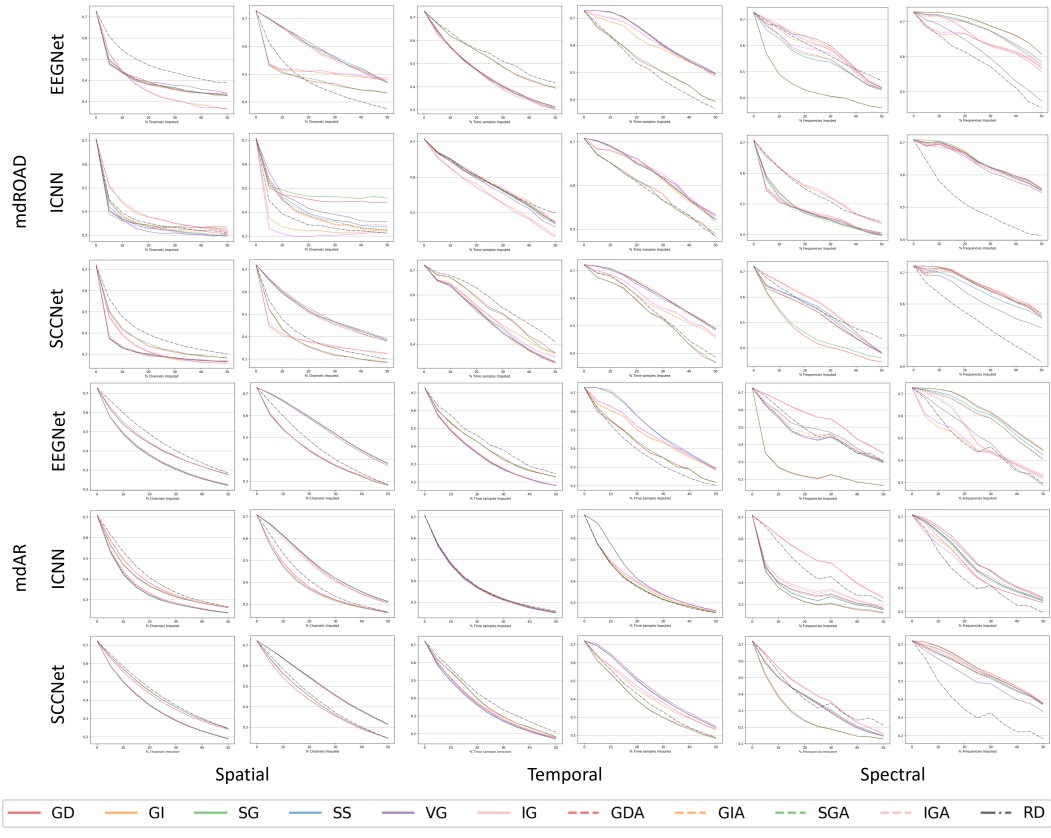

Figure 5: Extend performance-masking ratio curves for SMR dataset. (Odd columns: MoRF, Even columns: LeRF)

## D    EXTENDED EXPERIMENTAL RESULTS

### D.1    EFFECTS OF DIFFERENT IMPUTATION METHOD IN MDROAD FRAMEWORK

To explore the effect of different stochastic processes in the temporal domain imputation of mdROAD framework, we conducted an extended experiment with EEGNet on the ERN dataset,

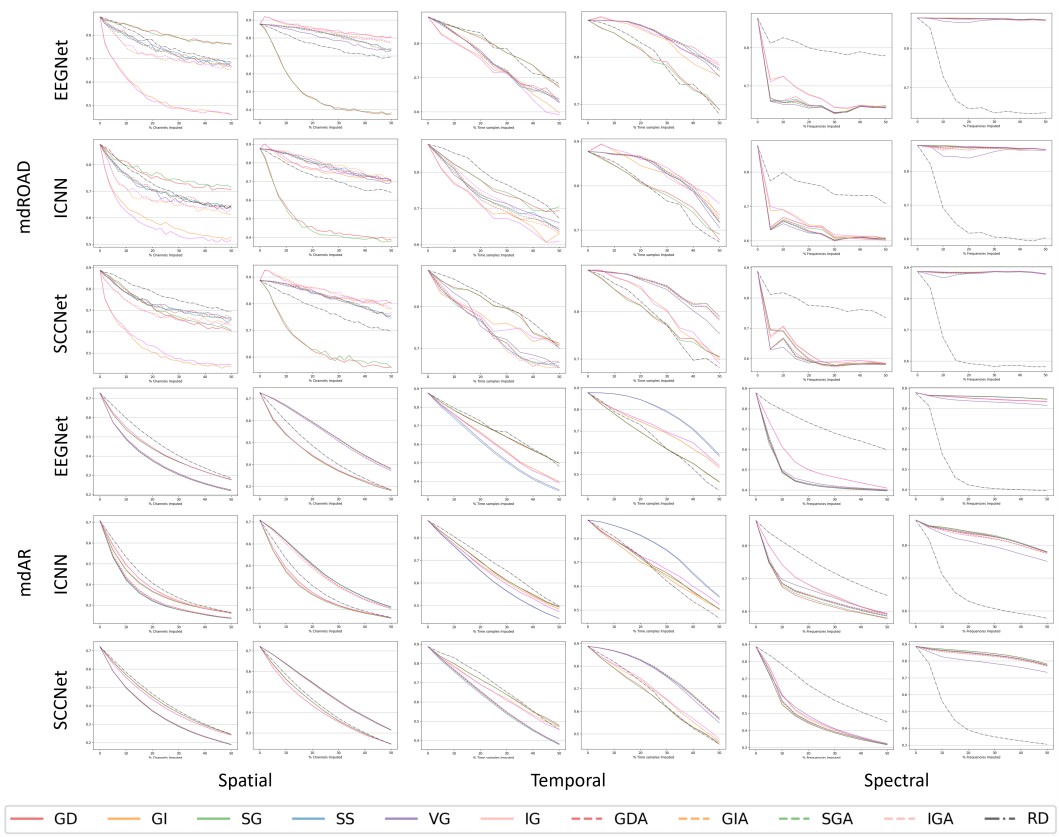

Figure 6: Extend performance-masking ratio curves for ERN dataset. (Odd columns: MoRF, Even columns: LeRF)

with models from 5 repeats. From the results, we can observe that using different stochastic processes does not greatly affect the faithfulness measurement, and IG is the most faithful explanation method in this dataset-model configuration. The results of the original mdROAD framework that averaged in three models using MFBB for imputation are provided in Table7.

Similarly, to explore the effect of different spectral domain imputation designs of the mdROAD framework, we conducted an extended experiment with EEGNet on the SSVEP dataset, with models from 5 repeats. In line with the discussion in Sections 5.1 and 5.3, explanation methods that preserve the sign of the saliency pattern are measured to be more faithful. In addition, explanations with absolute performed worse in metrics that consider the MoRF order with the unnatural imputation method.

### D.2 FAITHFULNESS EVALUATION RESULT AND RANKINGS

The unscaled faithfulness metrics are presented in Table 6, 7, and 8. For fair comparison, the intrinsic differences of classification performance between model structures or datasets and the nonparallel distribution of raw faithfulness measurement values should be considered. Firstly, to eliminate model and dataset variabilities, we clipped and scaled the raw accuracies to [chance level-original accuracy (0% information masked)] before computing the area-based metrics. The chance level datasets refer to are {0.25, 0.5, 0.2} for {SMR, ERN, SSVEP} dataset, respectively. The results from 5 differently random seeded set of models are averaged after this step.

For a comprehensible comparison of faithfulness between explanation methods, their ranking averaged across models are displayed in Fig. 8. The inconsistency in spectral domain comparing to the other two was discussed in Section 5.3.

Table 4: Comparison of faithfulness scores for feature attribution methods using different imputation method in mdROAD framework with spectral domain, EEGNet on the ERN dataset. Higher values indicate greater faithfulness. Highlighted cells represent the "most faithful" method within each column.

| method | mdROAD, Temporal domain with EEGNet on ERN dataset | | | | | | | | |
| | **MFBB**, $\rho$: 0.538±0.162 | | | **Gaussian Process**, $\rho$:0.64±0.164 | | | **Uniform Distribution**, $\rho$:0.721±0.168 | | |
| | AOC | ABC | AUC | AOC | ABC | AUC | AOC | ABC | AUC |
|---|---|---|---|---|---|---|---|---|---|
| GD | 0.221±0.019 | 0.013±0.010 | 0.784±0.015 | 0.220±0.017 | 0.019±0.013 | 0.792±0.017 | 0.207±0.023 | 0.026±0.027 | 0.813±0.027 |
| GI | 0.282±0.022 | 0.111±0.008 | 0.829±0.016 | 0.305±0.019 | 0.153±0.013 | 0.847±0.017 | 0.284±0.032 | 0.146±0.021 | 0.862±0.019 |
| SG | 0.220±0.019 | 0.012±0.010 | 0.783±0.019 | 0.220±0.015 | 0.018±0.011 | 0.792±0.024 | 0.206±0.022 | 0.025±0.026 | 0.813±0.028 |
| SS | 0.265±0.014 | 0.106±0.017 | 0.841±0.017 | 0.287±0.018 | 0.137±0.011 | 0.850±0.021 | 0.257±0.025 | 0.115±0.016 | 0.859±0.023 |
| VG | 0.263±0.005 | 0.100±0.021 | 0.836±0.022 | 0.289±0.010 | 0.140±0.016 | 0.850±0.022 | 0.256±0.026 | 0.113±0.018 | 0.858±0.025 |
| IG | 0.290±0.023 | 0.128±0.009 | 0.838±0.014 | 0.322±0.030 | 0.177±0.016 | 0.855±0.017 | 0.295±0.038 | 0.162±0.026 | 0.866±0.021 |
| GDA | 0.263±0.015 | 0.103±0.014 | 0.840±0.017 | 0.289±0.017 | 0.141±0.011 | 0.851±0.020 | 0.256±0.026 | 0.116±0.015 | 0.860±0.022 |
| GIA | 0.262±0.007 | 0.104±0.011 | 0.843±0.017 | 0.290±0.015 | 0.145±0.009 | 0.856±0.020 | 0.258±0.025 | 0.120±0.014 | 0.862±0.021 |
| SGA | 0.264±0.015 | 0.103±0.014 | 0.839±0.017 | 0.288±0.018 | 0.140±0.006 | 0.852±0.019 | 0.257±0.025 | 0.116±0.013 | 0.860±0.022 |
| IGA | 0.264±0.008 | 0.107±0.012 | 0.843±0.017 | 0.293±0.016 | 0.146±0.011 | 0.853±0.021 | 0.257±0.024 | 0.119±0.014 | 0.862±0.022 |
| RD | 0.220±0.015 | 0.013±0.005 | 0.788±0.019 | 0.215±0.018 | 0.012±0.002 | 0.794±0.022 | 0.198±0.022 | 0.011±0.007 | 0.810±0.029 |

Table 5: Comparison of faithfulness scores for feature attribution methods using different stochastic processes in mdROAD framework with temporal domain, EEGNet on the SSVEP dataset. Higher values indicate greater faithfulness. Highlighted cells represent the "most faithful" method within each column.

| method | mdROAD, Spectral domain with EEGNet on SSVEP dataset | | | | | |
| | **Polynomial of degree 3** $\rho$: 0.318±0.340 | | | **Linear interpolation** $\rho$: 0.473±0.221 | | |
| | AOC | ABC | AUC | AOC | ABC | AUC |
|---|---|---|---|---|---|---|
| GD | 0.643±0.021 | 0.284±0.009 | 0.641±0.013 | 0.626±0.087 | 0.259±0.057 | 0.633±0.032 |
| GI | 0.617±0.021 | 0.242±0.013 | 0.625±0.015 | 0.617±0.092 | 0.247±0.076 | 0.630±0.017 |
| SG | 0.644±0.021 | 0.286±0.009 | 0.642±0.014 | 0.623±0.089 | 0.262±0.074 | 0.639±0.016 |
| SS | 0.603±0.011 | 0.230±0.010 | 0.627±0.013 | 0.546±0.127 | 0.160±0.118 | 0.614±0.013 |
| VG | 0.650±0.007 | 0.217±0.015 | 0.567±0.016 | 0.532±0.134 | 0.100±0.113 | 0.551±0.013 |
| IG | 0.618±0.018 | 0.245±0.010 | 0.627±0.014 | 0.612±0.094 | 0.242±0.079 | 0.630±0.018 |
| GDA | 0.597±0.012 | 0.209±0.007 | 0.612±0.012 | 0.492±0.020 | 0.093±0.010 | 0.596±0.014 |
| GIA | 0.587±0.015 | 0.206±0.008 | 0.619±0.016 | 0.509±0.020 | 0.125±0.007 | 0.615±0.014 |
| SGA | 0.597±0.011 | 0.210±0.008 | 0.613±0.013 | 0.494±0.020 | 0.097±0.008 | 0.599±0.014 |
| IGA | 0.591±0.016 | 0.213±0.007 | 0.622±0.016 | 0.515±0.020 | 0.130±0.006 | 0.614±0.015 |
| RD | 0.493±0.022 | 0.000±0.000 | 0.404±0.013 | 0.542±0.109 | 0.049±0.080 | 0.483±0.017 |

Table 6: Complete faithfulness measurements on SMR dataset.

| method | Spatial mdROAD AOC | Spatial mdROAD ABC | Spatial mdROAD AUC | Spatial mdAR AOC | Spatial mdAR ABC | Spatial mdAR AUC | Temporal mdROAD AOC | Temporal mdROAD ABC | Temporal mdROAD AUC | Temporal mdAR AOC | Temporal mdAR ABC | Temporal mdAR AUC | Spectral mdROAD AOC | Spectral mdROAD ABC | Spectral mdROAD AUC | Spectral mdAR AOC | Spectral mdAR ABC | Spectral mdAR AUC |
|---|---|---|---|---|---|---|---|---|---|---|---|---|---|---|---|---|---|---|
| GD | .663±.131 | .297±.185 | .408±.170 | .702±.273 | .009±.007 | .252±.278 | .278±.144 | .085±.083 | .452±.210 | .770±.286 | .031±.046 | .224±.279 | .726±.108 | .770±.146 | .783±.118 | .848±.253 | .645±.221 | .650±.211 |
| GI | .735±.146 | .313±.290 | .328±.239 | .699±.271 | .007±.009 | .250±.275 | .521±.250 | .530±.231 | .786±.121 | .811±.289 | .288±.223 | .314±.281 | .270±.123 | .315±.107 | .701±.056 | .480±.212 | .129±.182 | .448±.247 |
| SG | .658±.127 | .318±.210 | .427±.176 | .703±.274 | .008±.007 | .251±.279 | .282±.149 | .083±.088 | .455±.201 | .770±.288 | .040±.055 | .229±.279 | .715±.109 | .762±.136 | .785±.108 | .843±.258 | .613±.236 | .622±.232 |
| SS | .763±.171 | .615±.273 | .583±.238 | .790±.279 | .634±.209 | .483±.230 | .516±.310 | .635±.291 | .919±.046 | .826±.291 | .539±.289 | .418±.267 | .453±.163 | .494±.124 | .733±.097 | .687±.254 | .401±.129 | .588±.227 |
| VG | .759±.205 | .645±.221 | .607±.214 | .785±.278 | .603±.190 | .472±.230 | .510±.284 | .630±.253 | .919±.054 | .823±.291 | .521±.276 | .412±.266 | .405±.224 | .328±.223 | .568±.081 | .660±.259 | .271±.168 | .473±.209 |
| IG | .734±.138 | .322±.299 | .320±.260 | .699±.270 | .008±.011 | .251±.275 | .557±.251 | .612±.233 | .844±.111 | .815±.289 | .343±.245 | .337±.281 | .271±.113 | .313±.096 | .699±.057 | .479±.212 | .129±.185 | .447±.253 |
| GDA | .750±.175 | .584±.315 | .568±.252 | .788±.277 | .623±.214 | .480±.234 | .503±.311 | .615±.282 | .909±.047 | .784±.290 | .376±.252 | .296±.286 | .464±.167 | .461±.151 | .737±.090 | .680±.252 | .384±.128 | .579±.217 |
| GIA | .751±.172 | .573±.301 | .558±.249 | .785±.276 | .602±.212 | .472±.237 | .507±.327 | .626±.298 | .918±.045 | .813±.289 | .214±.236 | .354±.278 | .424±.192 | .523±.103 | .723±.059 | .662±.246 | .279±.148 | .480±.226 |
| SGA | .755±.170 | .581±.298 | .562±.250 | .789±.279 | .631±.212 | .483±.230 | .509±.311 | .622±.286 | .910±.043 | .785±.292 | .411±.262 | .300±.284 | .469±.164 | .482±.137 | .752±.081 | .687±.253 | .388±.128 | .574±.226 |
| IGA | .766±.170 | .594±.248 | .566±.229 | .783±.277 | .589±.216 | .467±.239 | .508±.318 | .629±.286 | .921±.048 | .815±.289 | .012±.026 | .368±.278 | .446±.182 | .482±.137 | .726±.061 | .672±.241 | .296±.142 | .488±.225 |
| RD | .465±.245 | .011±.007 | .312±.144 | .649±.258 | .004±.003 | .305±.261 | .175±.114 | .014±.010 | .441±.202 | .731±.283 | .012±.026 | .226±.296 | .290±.103 | .005±.008 | .219±.159 | .585±.224 | .001±.002 | .259±.270 |

Table 7: Complete faithfulness measurements on ERN dataset.

| method | Spatial mdROAD AOC | Spatial mdROAD ABC | Spatial mdROAD AUC | Spatial mdAR AOC | Spatial mdAR ABC | Spatial mdAR AUC | Temporal mdROAD AOC | Temporal mdROAD ABC | Temporal mdROAD AUC | Temporal mdAR AOC | Temporal mdAR ABC | Temporal mdAR AUC | Spectral mdROAD AOC | Spectral mdROAD ABC | Spectral mdROAD AUC | Spectral mdAR AOC | Spectral mdAR ABC | Spectral mdAR AUC |
|---|---|---|---|---|---|---|---|---|---|---|---|---|---|---|---|---|---|---|
| GD | .193±.158 | .000±.001 | .149±.132 | .381±.185 | .003±.002 | .386±.149 | .258±.159 | .104±.102 | .394±.156 | .376±.236 | .062±.081 | .325±.144 | .809±.086 | .773±.087 | .887±.081 | .801±.236 | .790±.199 | .900±.080 |
| GI | .831±.109 | .764±.187 | .837±.110 | .311±.175 | .016±.008 | .455±.140 | .556±.217 | .548±.232 | .633±.144 | .535±.223 | .205±.149 | .418±.143 | .752±.112 | .730±.100 | .885±.079 | .704±.267 | .701±.211 | .880±.075 |
| SG | .181±.144 | .000±.001 | .153±.142 | .382±.184 | .003±.002 | .386±.149 | .255±.154 | .094±.104 | .383±.159 | .373±.233 | .053±.074 | .318±.137 | .816±.089 | .779±.089 | .888±.081 | .796±.244 | .788±.203 | .902±.078 |
| SS | .298±.106 | .270±.059 | .827±.077 | .664±.135 | .828±.096 | .842±.136 | .531±.094 | .638±.082 | .778±.149 | .727±.176 | .765±.139 | .838±.153 | .839±.103 | .794±.100 | .886±.080 | .778±.251 | .765±.209 | .893±.081 |
| VG | .294±.110 | .232±.086 | .794±.088 | .600±.141 | .706±.088 | .793±.141 | .499±.102 | .568±.084 | .732±.150 | .695±.178 | .723±.137 | .819±.155 | .853±.104 | .780±.107 | .861±.089 | .761±.277 | .667±.228 | .806±.112 |
| IG | .849±.093 | .790±.161 | .846±.108 | .311±.174 | .018±.007 | .457±.140 | .593±.228 | .630±.264 | .686±.155 | .561±.220 | .286±.143 | .480±.142 | .750±.110 | .729±.100 | .885±.079 | .701±.268 | .701±.213 | .883±.075 |
| GDA | .294±.098 | .257±.058 | .819±.081 | .666±.135 | .832±.095 | .844±.136 | .540±.132 | .656±.126 | .776±.143 | .488±.247 | .292±.319 | .502±.275 | .842±.100 | .793±.099 | .883±.082 | .778±.255 | .759±.217 | .886±.085 |
| GIA | .402±.101 | .402±.059 | .863±.085 | .552±.148 | .644±.087 | .780±.138 | .536±.113 | .633±.075 | .793±.139 | .606±.207 | .404±.260 | .565±.240 | .835±.105 | .788±.102 | .883±.080 | .774±.258 | .739±.211 | .868±.078 |
| SGA | .296±.101 | .263±.046 | .822±.082 | .667±.135 | .833±.096 | .844±.136 | .535±.110 | .633±.075 | .768±.144 | .483±.249 | .285±.323 | .498±.275 | .844±.101 | .796±.099 | .885±.080 | .777±.257 | .761±.214 | .889±.083 |
| IGA | .393±.101 | .387±.068 | .856±.093 | .548±.147 | .632±.089 | .774±.141 | .514±.087 | .619±.100 | .775±.163 | .623±.199 | .465±.206 | .614±.207 | .836±.104 | .789±.101 | .884±.080 | .775±.259 | .741±.212 | .870±.078 |
| RD | .206±.114 | .023±.023 | .658±.088 | .224±.172 | .003±.002 | .511±.137 | .194±.107 | .042±.037 | .376±.152 | .270±.174 | .012±.018 | .333±.142 | .173±.112 | .007±.004 | .109±.081 | .287±.197 | .000±.000 | .139±.161 |

Table 8: Complete faithfulness measurements on SSVEP dataset.

| method | Spatial mdROAD AOC | Spatial mdROAD ABC | Spatial mdROAD AUC | Spatial mdAR AOC | Spatial mdAR ABC | Spatial mdAR AUC | Temporal mdROAD AOC | Temporal mdROAD ABC | Temporal mdROAD AUC | Temporal mdAR AOC | Temporal mdAR ABC | Temporal mdAR AUC | Spectral mdROAD AOC | Spectral mdROAD ABC | Spectral mdROAD AUC | Spectral mdAR AOC | Spectral mdAR ABC | Spectral mdAR AUC |
|---|---|---|---|---|---|---|---|---|---|---|---|---|---|---|---|---|---|---|
| GD | .299±.222 | .004±.005 | .255±.165 | .291±.223 | .014±.014 | .382±.242 | .351±.208 | .046±.027 | .387±.243 | .461±.333 | .022±.024 | .362±.250 | .664±.068 | .757±.126 | .815±.111 | .762±.195 | .738±.198 | .788±.123 |
| GI | .675±.153 | .501±.246 | .526±.246 | .376±.240 | .102±.055 | .391±.247 | .396±.118 | .248±.185 | .486±.344 | .462±.334 | .125±.108 | .399±.254 | .575±.108 | .653±.127 | .786±.095 | .605±.180 | .492±.154 | .702±.154 |
| SG | .289±.219 | .005±.010 | .255±.158 | .291±.223 | .014±.012 | .382±.244 | .338±.206 | .044±.026 | .390±.240 | .462±.330 | .024±.022 | .364±.250 | .677±.075 | .776±.120 | .822±.108 | .751±.182 | .710±.183 | .771±.138 |
| SS | .730±.155 | .780±.102 | .761±.129 | .587±.284 | .724±.208 | .771±.173 | .533±.269 | .533±.290 | .705±.179 | .610±.284 | .562±.239 | .589±.300 | .687±.208 | .744±.174 | .785±.106 | .231±.092 | .098±.107 | .593±.233 |
| VG | .730±.166 | .771±.124 | .752±.119 | .572±.286 | .688±.203 | .751±.168 | .531±.275 | .527±.318 | .700±.181 | .594±.273 | .526±.269 | .581±.305 | .789±.133 | .696±.169 | .662±.095 | .162±.090 | .016±.022 | .367±.262 |
| IG | .728±.125 | .615±.155 | .594±.225 | .383±.245 | .123±.061 | .404±.243 | .507±.196 | .414±.120 | .621±.207 | .525±.299 | .214±.091 | .459±.269 | .589±.110 | .673±.130 | .794±.096 | .592±.182 | .481±.140 | .704±.155 |
| GDA | .725±.154 | .761±.120 | .746±.137 | .584±.287 | .718±.214 | .768±.171 | .528±.272 | .520±.309 | .698±.178 | .561±.280 | .565±.236 | .630±.291 | .680±.218 | .694±.183 | .745±.103 | .236±.237 | .055±.085 | .508±.237 |
| GIA | .716±.171 | .732±.118 | .724±.136 | .572±.286 | .688±.215 | .751±.172 | .523±.270 | .515±.299 | .699±.177 | .544±.273 | .507±.259 | .610±.297 | .648±.221 | .705±.132 | .779±.089 | .254±.086 | .040±.058 | .469±.238 |
| SGA | .735±.159 | .770±.101 | .745±.126 | .585±.283 | .720±.205 | .769±.172 | .537±.281 | .532±.311 | .700±.178 | .558±.277 | .559±.240 | .629±.292 | .678±.214 | .701±.187 | .752±.104 | .240±.091 | .070±.092 | .518±.255 |
| IGA | .715±.167 | .723±.109 | .717±.136 | .570±.287 | .679±.214 | .745±.172 | .526±.269 | .499±.317 | .678±.191 | .535±.260 | .475±.297 | .599±.311 | .653±.220 | .717±.140 | .787±.089 | .272±.088 | .064±.081 | .492±.245 |
| RD | .343±.160 | .017±.014 | .332±.155 | .257±.204 | .011±.008 | .422±.235 | .317±.159 | .049±.038 | .424±.210 | .432±.333 | .046±.031 | .421±.254 | .200±.126 | .000±.000 | .170±.100 | .398±.133 | .026±.034 | .393±.115 |

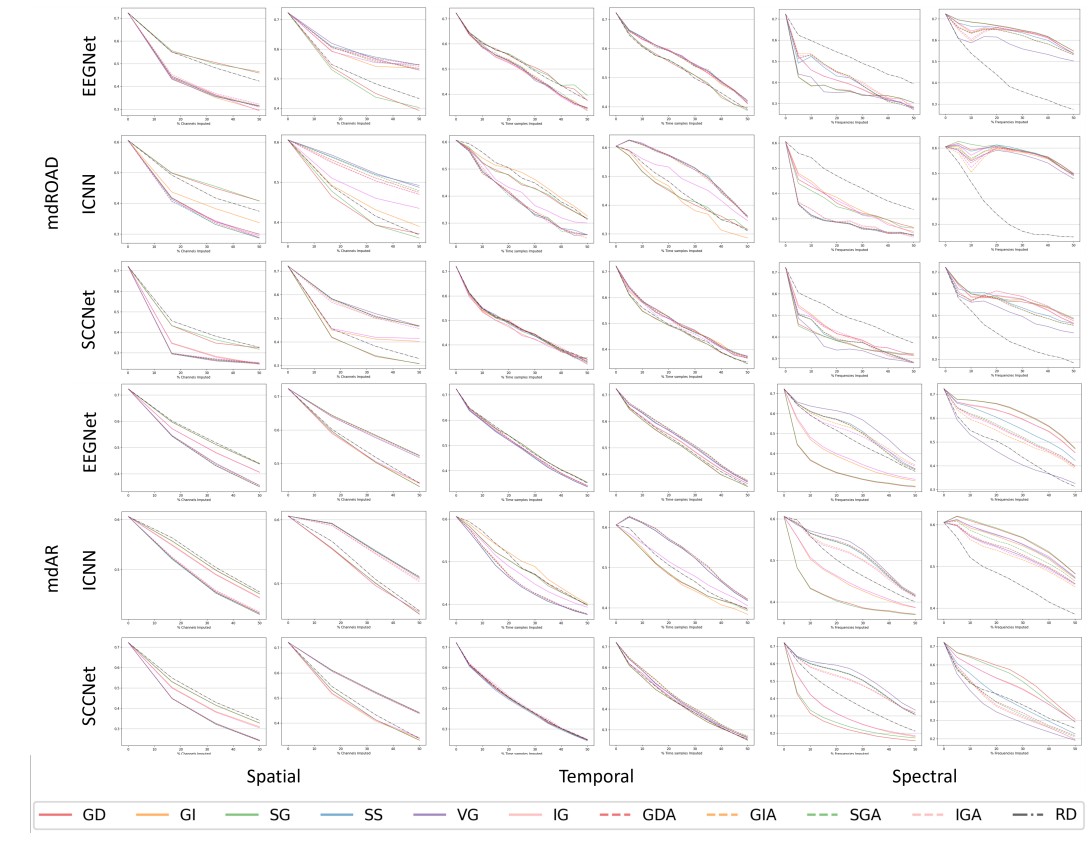

Figure 7: Extend performance-masking ratio curves for SSVEP dataset. (Odd columns: MoRF, Even columns: LeRF)

## E  EXTENDED EXAMPLE OF FEATURE ATTRIBUTION RESULTS

In Fig 9, 10 and 11, we present the class-wise feature attribution results from the best-performing model in one repeat, for a selected subject from each dataset. It is important to note that our imputation experiments were conducted on a per-input basis and were devoid of class information.

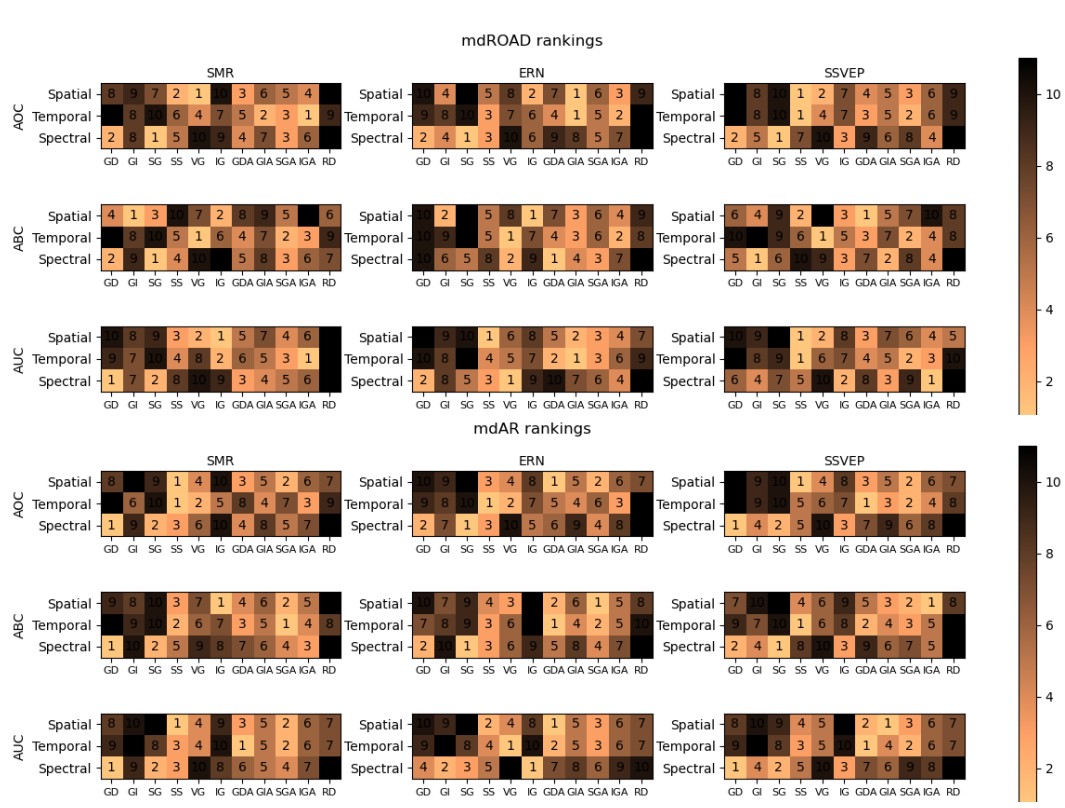

Figure 8: Faithfulness ranking across explanation methods by area-centic metrics.

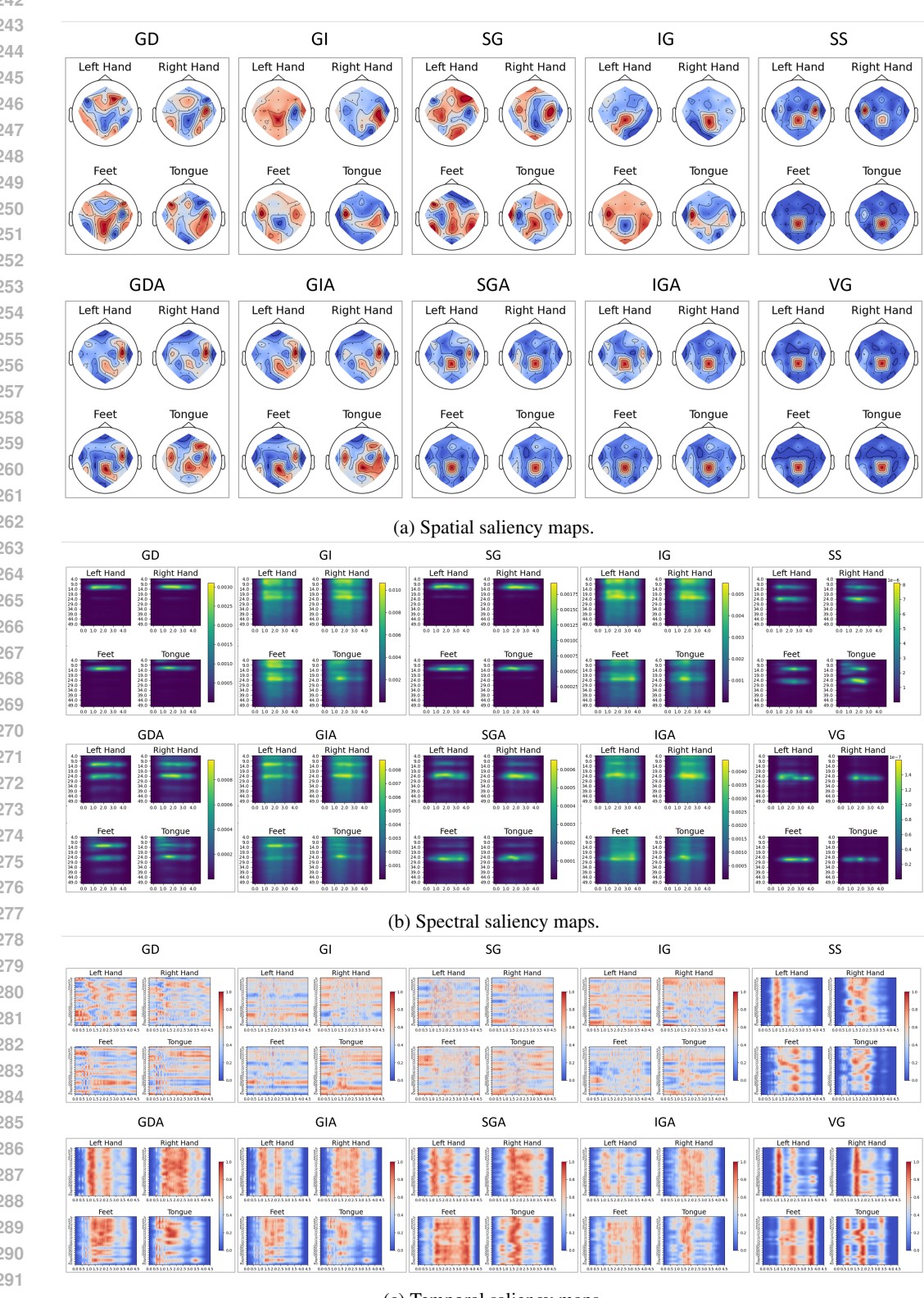

(a) Spatial saliency maps.

(b) Spectral saliency maps.

(c) Temporal saliency maps.

Figure 9: Example of SMR dataset feature attributions. The examples are generated with EEGNet from one out of 5 repeats on subject 3.

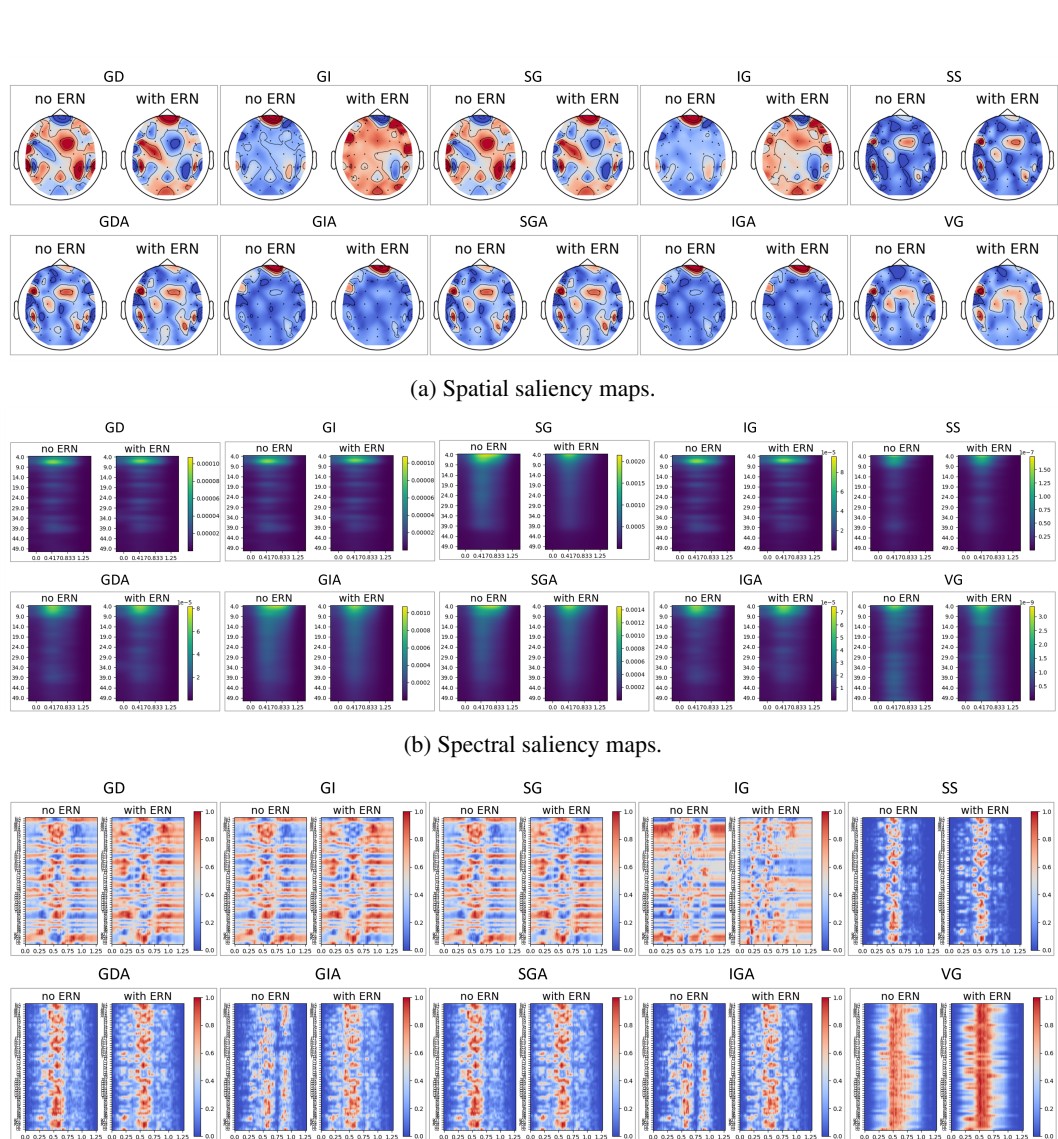

(a) Spatial saliency maps.

(b) Spectral saliency maps.

(c) Temporal saliency maps.

Figure 10: Example of ERN dataset feature attributions. The examples are generated with EEGNet from one out of 5 repeats on subject 7.

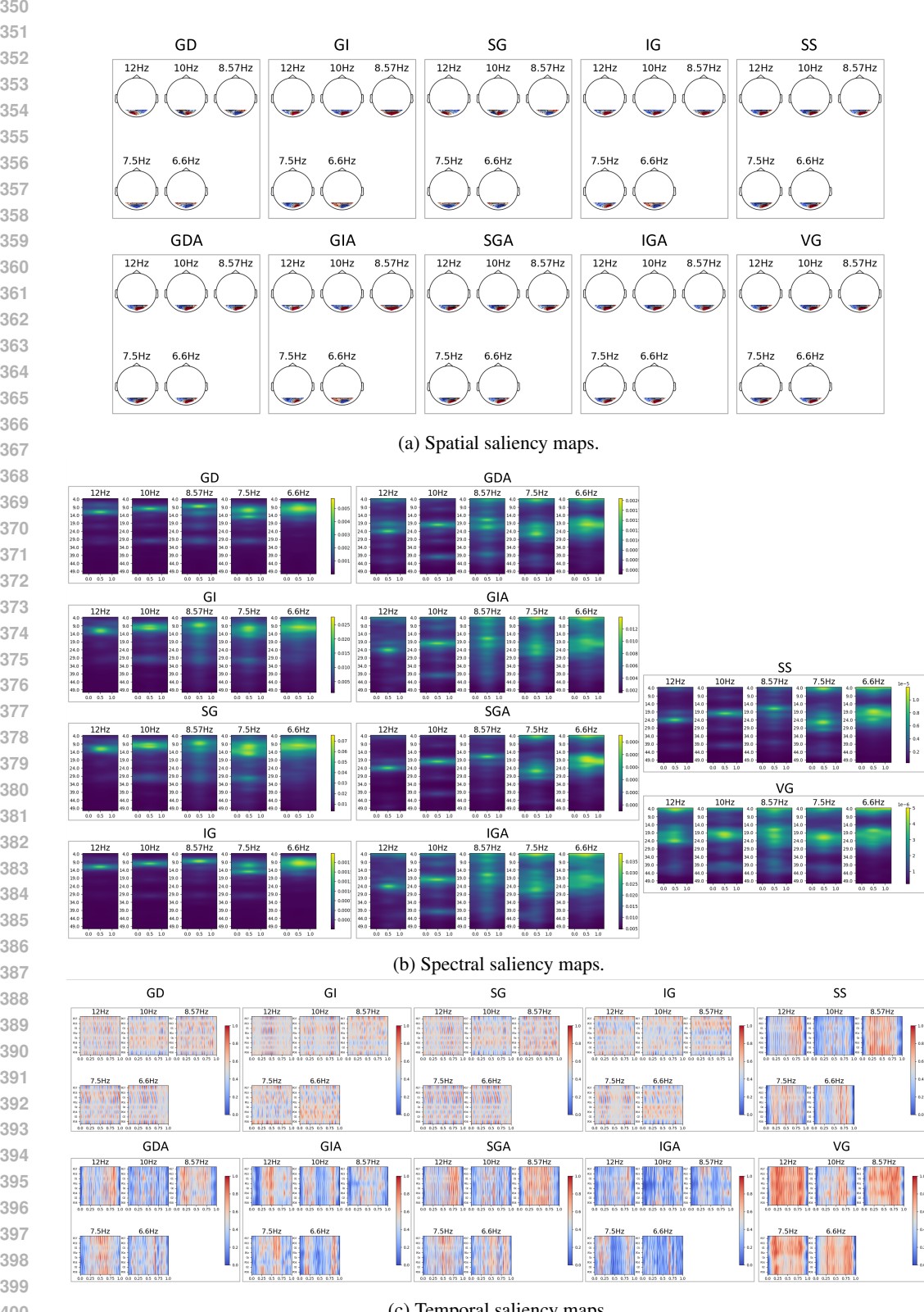

(a) Spatial saliency maps.

(b) Spectral saliency maps.

(c) Temporal saliency maps.

Figure 11: Example of SSVEP dataset feature attributions. The examples are generated with EEG-Net from one out of 5 repeats on subject 2.

