# OpenReview forum: "AIM: Adversarial Information Masking for Evaluating EEG-DL Interpretations"
_ICLR.cc/2025/Conference — ICLR 2025 Conference Withdrawn Submission_

### Official Review · Reviewer_cFhK · 2024-10-30

**Soundness:** 2
**Presentation:** 1
**Contribution:** 2
**Rating:** 3
**Confidence:** 3

**Summary:**

The manuscript aims to contribute to the evaluation of interpretations of deep learning models applied to EEG. It does so by identifying issues with traditional adversarial robustness evaluation for EEG and proposing alternative information masking methods to evaluate the faithfulness of feature attribution methods. Their framework is able to differentiate between attribution methods on spatial, temporal, and spectral domains and thereby seem to generate useful findings for the field. Such progress is valuable as explainability of neural networks in EEG data is not well-studied.

**Strengths:**

-	Addresses a real gap in the EEG/DL field
-	Seems to generate useful findings
-	Three EEG domains are considered

**Weaknesses:**

1.Writing: Unfortunately, the level of the writing of the manuscript is poor. Especially the first half of the paper, outlining the motivations, prior literature, and outline of the paper, is difficult to follow. The paper could really benefit from another thorough round of editing as the many grammatical errors lead to semantic ambiguity. A few examples that I am unable to understand:
a. L218-219: ‘ability to exert data distribution’
b. L220-221:‘computationally exhaustive while remain biased or uncontrollable’.
c. L271-273: "(…), whose value are concluded to reflect certain series trend.”
Also the experiments are hard to follow and it is difficult to assess the contributions of this work.

2. Evaluation: Is it possible to perform some form of cross-framework comparison? It is difficult to understand the advantage of the proposed framework over existing ones. For example, why did the authors choose not to analyze where and why frameworks agree or disagree? Would synthetic data enable a comparison between frameworks? Understanding both the advantages and disadvantages of this new framework would be very valuable.

Minor:
-	It might be useful to briefly describe what the the proposed metrics (AOC, ABC) conceptionally mean and how they differ.
-	Recent work on EEG-DL concerns the use of large amounts of resting state data for clinical predictions (e.g. [1-3]). Do the authors believe the proposed framework could be relevant for such work as well? Understandably, task-based explanations are easier to verify and interpret. However, the datasets used by the authors are small, while it may be argued that deep learning models may be particularly interesting in case of larger datasets, which to my knowledge tend to be resting-state.

References:
1.	https://openreview.net/forum?id=QzTpTRVtrP
2.	https://arxiv.org/abs/2305.10351
3.	https://arxiv.org/abs/2409.07480

**Questions:**

- Improve clarity and style of writing
- Be more concise in your contributions
- Evaluate your framework with respect to other frameworks
- Give more context regarding the EEG experiments and show more example attributions

---

> ### Author Response · Authors · 2024-11-27
> **Response to Reviewer cFhK**
>
> We sincerely thank the reviewer for the thoughtful review and constructive comments, and apologize for the delay in our rebuttal, as we aimed to provide the reviewer with a thoroughly refined version. We appreciate the valuable insights provided and would like to respond to your feedback as follows. Additionally, we have made major revisions based on the reviews, particularly to: 1) enhance the clarity of our study’s premise, 2) emphasize its contributions, and 3) revise the metric computation to facilitate a more direct analysis. We look forward to your continued evaluation of our work.
>
> > (Weaknesses) Writing
>
> We have conducted a thorough review of the manuscript to identify and address grammatical errors, clarity issues, and writing style deficiencies. Significant revisions have been made throughout the document to enhance overall readability. We invite you to review the revised version, where the modified sections are highlighted in red.
>
> > (Weaknesses) Give more context regarding the EEG experiments
>
> We acknowledge the importance of providing additional context for the EEG experiments. While we aim to present a comprehensive understanding of our methodology and findings, we plan to include more detailed examples of attributions in the revised manuscript. We are considering the possibility of extracting the imputation function to make the explanation clearer or tabulating the target-imputation functions for each domain for easier comparison.
>
> > (Weaknesses) Cross-framework comparison
>
> We completely agree that a comparison with other frameworks would be beneficial and could yield valuable insights. However, as our findings suggest, the evaluated faithfulness of each explanation method is closely tied to the specific characteristics of the dataset being analyzed. This variability makes direct comparisons challenging, but we recognize the potential for future research to delve deeper into this aspect.
>
> > (Weaknesses) It might be useful to briefly describe what the proposed metrics (AOC, ABC) conceptually mean and how they differ.
>
> We appreciate your suggestion to improve clarity concerning the faithfulness metrics. In response, we have revised Section 4.3, adding the following text to the end of the paragraph on Lines 376-377:
> - *Higher measurements indicate that the curves’ behavior aligns more closely with expectations, reflecting greater faithfulness. An illustrated example of the metrics is displayed in Figure 1b.*
>
> > (Weaknesses) Do the authors believe the proposed framework could be relevant for recent interests in large amounts of data, for example, resting state EEG?
>
> Thank you for highlighting recent research that focuses on large volumes of non-task-related EEG data. Our information masking methods are designed to be input-specific, as they depend on feature attribution explanations for each specific input. As long as time-series-like saliency maps can be generated to represent all features in the input, our framework remains applicable. We are confident that our work will continue to play a crucial role in advancing the interpretability of EEG deep learning models, especially as they are applied to increasingly large and intricate datasets.
>
> References:
> 1. https://pmc.ncbi.nlm.nih.gov/articles/PMC8870584/
> 2. https://www.mdpi.com/2079-9292/13/1/186
> 3. https://journals.sagepub.com/doi/full/10.1177/15500594211063662
>
> > (Question) Be more concise in your contributions
>
> We have summarized our contributions more concisely as follows:
> - We expand the leading in-distribution information masking method, Remove and Debias, to accommodate multiple domains, including spatial, temporal, and spectral dimensions.
>
> - We introduce an adversarial information masking (AIM) approach to circumvent issues related to hand-crafted distribution selection and to enhance in-distribution information masking for multivariate time series data.
>
> - We assess the effectiveness of in-distribution information masking through a novel Multi-Domain Adversarial Robustness (mdAR) framework that includes new normalized faithfulness metrics and an evaluation result consistency-based methodology for framework validation.
>
> - We demonstrate assessments of faithfulness for existing post-hoc explanation methods and their limitations under specific conditions in the context of deep learning interpretation of human EEG data.
>
> > (Question) Show more example attributions
>
> We acknowledge the benefit of providing additional examples of attributions. While we strive for a comprehensive presentation of our proposed framework,  our primary focus in this work is on enhancing the information masking techniques and establishing a robust quantitative comparison framework.
>
> We sincerely hope that our revisions and responses adequately address your concerns and contribute to the clarity and impact of our work. Thank you once again for your valuable and constructive feedback.

---

> > ### Author Response · Authors · 2024-12-02
> >
> > Thank you once again for your insightful feedback as a reviewer. We have thoughtfully considered your suggestions and made substantial revisions to the manuscript accordingly. Your perspective is truly appreciated, and we invite you to review our rebuttal along with the updated manuscript. Based on the positive feedback from other reviewers, we hope our revisions effectively address your concerns and offer the clarity required for you to reassess your evaluation. If you need any further clarification or wish to discuss anything, please feel free to reach out.

---

> > > ### Author Response · Authors · 2024-12-03
> > >
> > > We would be grateful for your feedback on the revised manuscript. Please let us know if there are any remaining concerns or if further clarification is needed.

---

### Official Review · Reviewer_YJwj · 2024-11-03

**Soundness:** 3
**Presentation:** 3
**Contribution:** 2
**Rating:** 3
**Confidence:** 4

**Summary:**

The paper addresses the problem of evaluation of post-hoc explanations in the context of models trained on EEG decoding tasks.
In particular, it focuses on the evaluation of faithfulness of explanations and proposes a novel framework involving multi-domain adversarial information masking (AIM) based on Multi-Domain Adversarial Robustness (mdAR), which overcomes some of the limitations of standard faithfulness evaluation approaches. The framework is validated on multiple model architectures and EEG datasets.

**Strengths:**

The paper addresses a relevant problem, namely evaluation of post-hoc explanations. The paper is original in the sense that it proposes two imputation techniques specifically tailored for multivariate EEG data. The proposals are based on the ROAD and AR frameworks as the  and carefully integrate the spatial, spectral and temporal dimension of multivariate EEG data. The overall originally and quality of the proposed approach is rather limited and specific to models trained for EEG analysis. It is unclear how to generalise the approach beyond this specific application domain.
The paper is well written and easy to understand. The experimental evaluation is ok, but could be more detailed and deep. Currently it is not clear how follows, e.g., from the results in Table 2 or 3.
Overall, the contribution is rather incremental and will probably be of interest / significance only to a limited (EEG) community.

**Weaknesses:**

The contributions of the paper are very specific and may be of interest to a limited community, mainly only researchers training and explaining NN for EEG analysis. There has been a lot of research on faithfulness evaluation of explanations. The proposed method represents an incremental contribution to this field. The experimental evaluation is not 100% convincing. It is unclear to me what follows from the evaluation results. Shall we only use some of the methods which perform well in Table 2 for the analysis of EEG-based explanations? Are the results consistent with other evaluation approaches? What are the consequences of the evaluation for the practioner?.
Currently, the paper reads to me as proposing yet another faithfulness evolution metric, here specifically for EEG analysis tasks. The overall originality of the contribution and relevance for the ICLR research community is rather limited. Therefore I recommend "reject".

**Questions:**

What follows from the evaluation results?
What are the consequences for the practitioner?

---

> ### Comment · Reviewer_YJwj · 2024-11-27
> **No rebuttal**
>
> I will keep my rating

---

> > ### Author Response · Authors · 2024-11-27
> > **Response to Reviewer YJwj**
> >
> > We sincerely thank the reviewer for the thoughtful review and constructive comments, and apologize for the delay in our rebuttal, as we aimed to provide the reviewer with a thoroughly refined version. We appreciate the valuable insights provided and would like to respond to your feedback as follows. Additionally, we have made major revisions based on the reviews, particularly to: 1) enhance the clarity of our study’s premise, 2) emphasize its contributions, and 3) revise the metric computation to facilitate a more direct analysis. We look forward to your continued evaluation of our work.
> >
> > > (Weaknesses) It is unclear to me what follows from the evaluation results. Shall we only use some of the methods which perform well in Table 2 for the analysis of EEG-based explanations? What are the consequences of the evaluation for the practitioner?
> >
> > > (Questions)  What follows from the evaluation results?
> >
> > We understand your interest in the practical implications of our evaluation results. Our aim is not to prescribe a definitive set of "best" or "must-use" explanation methods for EEG deep learning (EEG-DL) applications; rather, we propose a guiding framework for practitioners. This framework is designed to assist practitioners in selecting the explanation methods that are most appropriate for their specific datasets or in identifying methods that reliably highlight key features within certain domains.
> >
> > For instance, we emphasize that practitioners should ensure the preservation of the sign of the utilized explanation method when visualizing frequency domain features. This precaution mitigates the risk of displaying confounding patterns. Additionally, our framework provides a systematic method for evaluating whether a specific explanation technique can accurately represent the intended feature domain. The insights garnered from this approach are crucial for practitioners to make informed decisions in real-world brain-computer interface (BCI) applications, thereby enhancing the interpretability of EEG-DL models in both academic research and practical contexts.
> >
> > > (Weaknesses) It is unclear to me what follows from the evaluation results. Are the results consistent with other evaluation approaches?
> >
> > We acknowledge your concern regarding the consistency of our evaluation results with other evaluation methodologies. Our results obtained from the mdROAD and mdAR frameworks represent two distinct evaluation strategies, as illustrated in Figure 8. While these results may not be identical, the relative trends in faithfulness—both high and low—are discernible. We have elaborated on potential reasons for these discrepancies in Section 5.2.
> >
> > Furthermore, as indicated in Section 2.4, the existing literature related to EEG-DL demonstrates substantial room for improvement in terms of evaluation strategies and experimental materials. Consequently, we respectfully argue that strict alignment with prior research was not a requisite for the objectives of our study.
> >
> > We hope these clarifications adequately address your concerns and enhance the overall impact of our work. Thank you again for your insightful feedback.

---

> > > ### Author Response · Authors · 2024-12-02
> > >
> > > Thank you once again for your valuable feedback and dedicated service as a reviewer. We have carefully considered your input and made significant updates to the manuscript in response. We genuinely value your perspective and encourage you to review our rebuttal and the revised manuscript. Given the positive feedback from other reviewers, we hope our revisions address your concerns and provide the clarity needed to reconsider your assessment. Please don not hesitate to reach out if further clarification or discussion would be helpful.

---

> > > > ### Author Response · Authors · 2024-12-03
> > > >
> > > > We would be grateful for your feedback on the revised manuscript. Please let us know if there are any remaining concerns or if further clarification is needed.

---

### Official Review · Reviewer_kAcF · 2024-11-03

**Soundness:** 3
**Presentation:** 3
**Contribution:** 3
**Rating:** 8
**Confidence:** 3

**Summary:**

The authors present approaches on how to mask EEG input data in spatial, frequency and temporal domains. The aim of this masking is faithfulness evaluation of attribution maps. The present novel ideas for masking in the temporal domain, also with likely a novelty for the frequency domain. Besides conventional approximate in-distribution masking they also evaluate masking by copying from adversarially crafted samples. The present results for several networks and several attribution methods. They investigate the question whether the sign of attribution maps carries information and investigate an unexpected result in the frequency domain.

**Strengths:**

It is a reasonable application study about faithfulness evaluation for a particular field - which is an acceptable type of invention. A good set of experiments in three domains, also for multiple networks. They measure also consistency in the sense of rank correlations.

**Weaknesses:**

In particular for the temporal domain, but also for the frequency domain, trying out different non-adversarial masking methods would make the evaluation more interesting. For the temporal domain that would be different stochastic processes.

**Questions:**

What are the footnotes in Table 2 ?

It might be fair to cite https://www.nature.com/articles/s42256-023-00620-w .

---

> ### Author Response · Authors · 2024-11-27
> **Response to Reviewer kAcF**
>
> We sincerely thank the reviewer for the thoughtful review and constructive comments, and apologize for the delay in our rebuttal, as we aimed to provide the reviewer with a thoroughly refined version. We appreciate the valuable insights provided and would like to respond to your feedback as follows. Additionally, we have made major revisions based on the reviews, particularly to: 1) enhance the clarity of our study’s premise, 2) emphasize its contributions, and 3) revise the metric computation to facilitate a more direct analysis. We look forward to your continued evaluation of our work.
>
>
>
> > (Questions) Footnote in Table 2
>
> We apologize for the oversight regarding the lack of description in Table 2. We have added a corresponding explanation in the last line of the table's caption:
> - *Highlighted cells represent the "most faithful" method, and the superscripts indicate the top-3 highest faithfulness measurements within each column.*
>
> > (Questions) Add citation of relevant work https://www.nature.com/articles/s42256-023-00620-w
>
> We are grateful to the reviewer for suggesting this relevant citation to enrich our survey. We have included the work recommended in the introduction of Section 2.4, specifically in Lines 201-202:
> - *With the growing understanding of post-hoc explanations in computer vision, there has been a recent expansion into other fields exploring this topic (Turbé et al., 2023; Fang et al., 2024).*
>
> > (Weaknesses) In particular for the temporal domain, but also for the frequency domain, trying out different non-adversarial masking methods would make the evaluation more interesting. For the temporal domain, that would involve different stochastic processes.
>
> We appreciate the reviewer's valuable suggestion regarding the exploration of non-adversarial masking methods. In response, we have conducted additional experiments utilizing various non-adversarial techniques in both the temporal and frequency domains. The results can be found in Appendix D.1.
>
> We hope that these revisions address your comments satisfactorily and enhance the clarity and quality of our paper. Thank you once again for your constructive feedback.

---

> ### Comment · Reviewer_kAcF · 2024-11-29
> **Reviewer kAcF - an argument for accepting this paper**
>
> Disclaimer: I am not aware at the current time (end of Nov 2024) who could be the authors of this submission.
>
> @Authors: Thank you for adding the additional requested experiments.
>
> In my view, the paper is a domain-specific analysis of faithfulness of different methods. Yes, it is incremental. Reporting more in-depth for existing measures.
>
> I have read the weaknesses from the other reviewers.
>
> As for reviewer YJwj,
> I would argue that having no strong conclusion is valid if several methods yield similar performances. Seeing no clear difference is a valid result in science. If we as a community would downvote that, it would risk to nudge the field towards sensationalistic reporting.
>
> Yes, they could write a more clear conclusion, but I do not see this as justifying a reject (3) level. For me a level 3 reject is something with methodical flaws, very poor readability or lacking or simplistic experiments.
> Same holds when adding the domain specificity to the above argument. DL for EEG is an active field in the sciences.
> (side note, AUC, ABC, AOC are up to the model prediction on the original sample not novel metrics)
>
> As for reviewer cFhK:
> faithfulness is pretty much established in vision as a category of evaluation metrics for attribution methods, with various variants having been developed (Samek et al, 2017 "Evaluating the visualization of what a
> Deep Neural Network has learned", Arya et al 2019 "One Explanation Does Not Fit All...", Alvarez-Melis, 2018 "Towards Robust Interpretability with Self-Explaining Neural Networks" and many more). Therefore, I also found the readability of this paper okay.
>
> Overall, the reviewer after reading the other weaknesses still thinks that this paper is ok to be accepted.

---

> ### Comment · Reviewer_kAcF · 2024-11-29
> **updated score**
>
> 30 Nov: raised the score from 6 to 8.
> Did not raise my confidence due to lack of familiarity with EEG data.

---

> > ### Author Response · Authors · 2024-12-03
> >
> > Thank you, Reviewer kAcF, for your thoughtful comments and for advocating for the acceptance of our paper.

---

### Note · Authors · 2025-01-16

**Comment:**

The authors would like to express their sincere gratitude to the reviewers for their time and effort in evaluating our submission. Your constructive feedback has provided invaluable guidance for refining our manuscript. After careful consideration, we have decided to withdraw the paper to focus on further improving our work.

**Withdrawal Confirmation:**

I have read and agree with the venue's withdrawal policy on behalf of myself and my co-authors.